# River interlinking alters land-atmosphere feedback and changes the Indian summer monsoon

Tejasvi Chauhan [1], Anjana Devanand [2,3,4], Mathew Koll Roxy [5], Karumuri Ashok [6,7] & Subimal Ghosh [1,2] ✉

Massive river interlinking projects are proposed to offset observed increasing droughts and floods in India, the most populated country in the world. These projects involve water transfer from surplus to deficit river basins through reservoirs and canals without an in-depth understanding of the hydro-meteorological consequences. Here, we use causal delineation techniques, a coupled regional climate model, and multiple reanalysis datasets, and show that land-atmosphere feedbacks generate causal pathways between river basins in India. We further find that increased irrigation from the transferred water reduces mean rainfall in September by up to 12% in already water-stressed regions of India. We observe more drying in La Niña years compared to El Niño years. Reduced September precipitation can dry rivers post-mon-soon, augmenting water stress across the country and rendering interlinking dysfunctional. Our findings highlight the need for model-guided impact assessment studies of large-scale hydrological projects across the globe.

Large international rivers, such as the Ganga, the Brahmaputra, and the Indus, are central to the development of agriculture-dominated India, with a 1.4 billion population[1]. However, like river basins around the globe[2], Indian river basins are also under severe stress due to global climate change[3], massive population growth[3–5], increased uncontrolled human water use[6,7], and pollution[8,9]. Indian summer monsoon (ISM) from June to September is the primary source of water in Indian river basins, which accounts for almost 80% of the country's annual rainfall and governs the Gross Domestic Product (GDP)[10]. Over the last few decades, ISM has experienced a decline in the mean rainfall[11,12] and an increase in the intensity, occurrences, and spatial variability of extreme rainfall[13–16]. Such changing meteorological patterns have increased hydrologic extremes, floods, and droughts in India[13,17–22]. These hydrologic changes have augmented the water stress across the country, elevating the risk of disasters. As an adaptation measure to combat the increasing hydrologic extremes, India has planned massive interlinking projects on its rivers with a proposed budget of USD 168 billion[23–25]. The proposal[23,26] involves a network of canals with an approximate length of 15,000 km and 3000 reservoirs with a capacity to transfer 174 billion cubic meters of water each year from surplus to deficit basins and generate 34 million kilowatts of hydropower along with benefits[27] like flood control, drought mitigation, and navigation. The experiences of river interlinking in China showed the stabilization of groundwater[28]; however, such an ambitious plan may significantly impact the ecology[29] of the aquatic ecosystem and fish diversity[30]. Literature also shows that increased river regulation could increase the water foot print[7]. Hence, interlinking must be carefully designed to optimize the conflicting objectives of ecological sustainability and meeting water demands[31].

So far, no scientific studies have explored the possibility of feedback from the inter-basin water transfer to the water cycle. We hypothesize that the water transfer may impact the donor or adjacent

[1]Department of Civil Engineering, Indian Institute of Technology Bombay, Mumbai, India. [2]Interdisciplinary Program in Climate Studies, Indian Institute of Technology Bombay, Mumbai, India. [3]Australian Research Council Centre of Excellence for Climate Extremes, University of New South Wales, Sydney, NSW, Australia. [4]Climate Change Research Centre, University of New South Wales, Sydney, NSW, Australia. [5]Centre for Climate Change Research, Indian Institute of Tropical Meteorology, Ministry of Earth Sciences, Pune, India. [6]Centre for Earth, Ocean and Atmospheric Sciences, University of Hyderabad, Hyderabad, India. [7]Physical Science and Engineering, King Abdullah University of Science and Technology, Thuwal, Saudi Arabia. ✉e-mail: subimal@iitb.ac.in

basins through land-atmosphere feedback. Such possibilities could be high in the Indian region, where the land feedback to the atmosphere is also high[32,33]. In the present study, we test the hypothesis by developing a causal network between the atmosphere and land variables across river basins in India. We use Granger Causality (GC)[34], information theory-based transfer entropy (TE)[35,36] and a causal network learning algorithm, PCMCI[37] (see "Methods" for details), to generate causal networks between different hydrometeorological variables.

We use these approaches on the variables (ST1), soil moisture (SM), latent heat flux (LH), sensible heat flux (SH), precipitation (P),

relative humidity (R), wind speed (WS) (resultant of u-wind (U), v-wind (V)), incoming shortwave radiation (SR), and temperature (T) over the major river basins of India (Fig. 1a), Ganga (G, 808,334 km²), Godavari (Go, 302,063 km²), Mahanadi (M, 139,659 km²), Krishna (K, 254,743 km²), Narmada-Tapi (NT, 98,796 km², 65,145 km², respectively −two river basins taken together), and Cauvery (C, 85,624 km²). Since Narmada and Tapi basins are relatively small, we club them together and represent them as a single basin. Relative humidity, u-wind, and v-wind are taken at 850 hpa pressure level, and the remaining variables are near-surface. We first use 40 years (1980–2019) of daily reanalysis

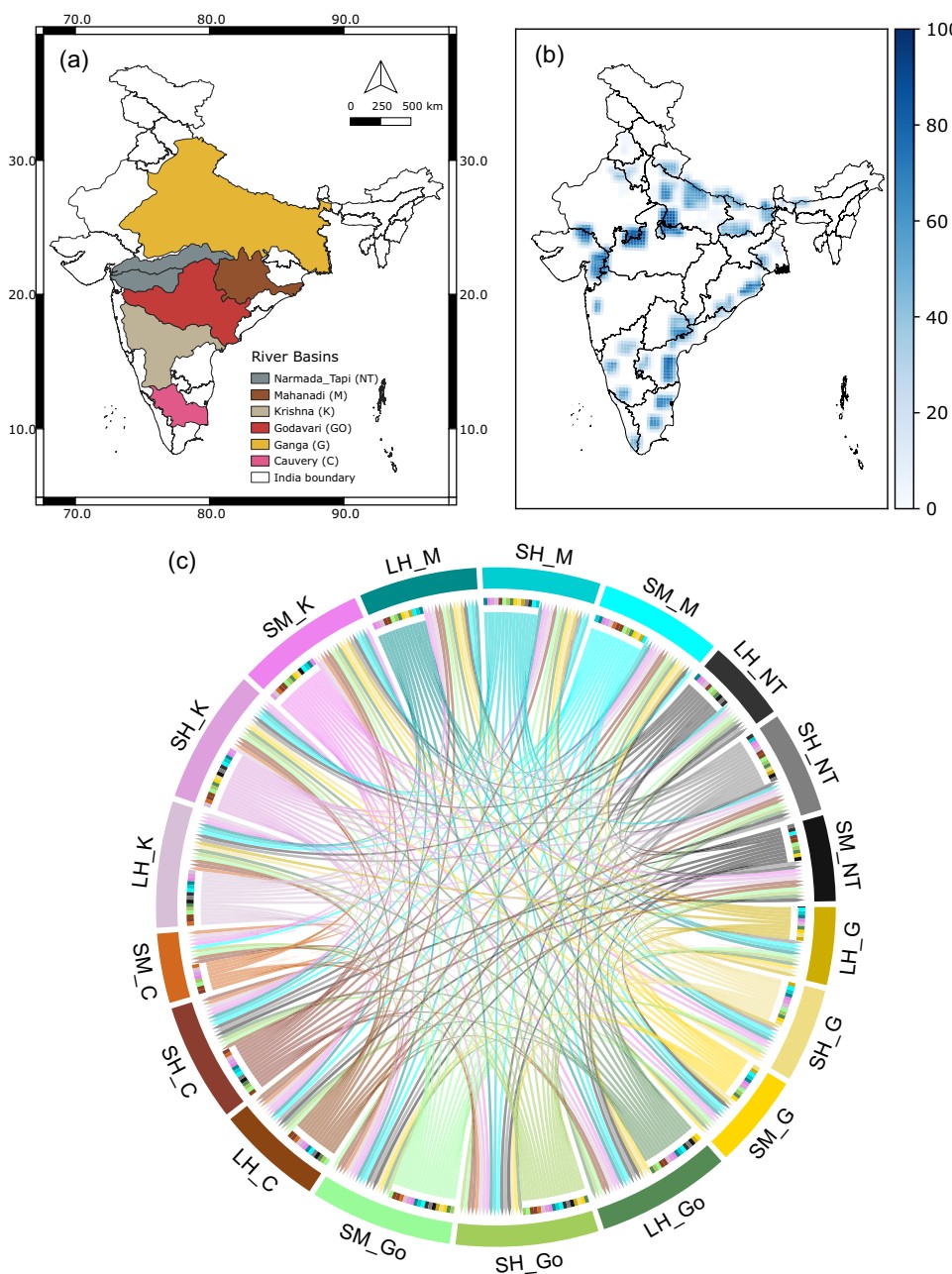

**Fig. 1 | Connections between land variables of different river basins. a** River Basins in India considered in the study. **b** Irrigated grid cells under river interlinking schemes showing change in percentage irrigated area from control run (CTL) to irrigation run (IRR) run (see "Methods") to increase irrigated area fraction to 80%. **c** Network between land variable across river basins generated using the algorithm PCMCI (ParCorr). Sectors are labeled as variable symbols (soil moisture-SM, latent heat flux-LH, sensible heat flux-SH) followed by the basin they belong to (Ganga (G, 808,334 km²), Godavari (Go, 302,063 km²), Mahanadi (M, 139,659 km²), Krishna (K,

254,743 km²), Narmada-Tapi (NT, 98,796 km², 65,145 km², respectively−two river basins taken together), and Cauvery (C, 85,624 km²)). Links are only shown if found statistically significant at 99% confidence and are colored same as the node they originate from. For example, link from LH_G to LH_M shows that there is a connection between latent heat fluxes from Ganga and Mahanadi basin. Ratio of incoming to outgoing links in Cauvery basin is very high compared to Ganga and Narmada-Tapi basin.

data from the European Center for Medium-Range Weather Forecast (ECMWF, ERA-5)[38]. Figure S1 presents the climatology of soil moisture, precipitation, and Evapotranspiration (ET, generated from latent heat flux) for different basins generated using ERA-5 variables. All the basins receive maximum precipitation during the Indian summer monsoon (also called southwest monsoon) from June to September[39]. The Cauvery basin also receives significant rainfall during October–December during the northeast monsoon[40] and has two peaks in annual precipitation. The soil moisture in Ganga, Godavari, Krishna Mahanadi, and Narmada-Tapi basin peaks during August and starts declining by the end of the summer monsoon season. The soil moisture in Cauvery peaks during late October, showing cumulative effects of rainfall from the Indian summer monsoon and northeast monsoon. Evapotranspiration in all basins increases during the start of the Indian summer monsoon and is highest during post-monsoon because moisture accumulated during monsoon gets evaporated by solar radiation.

After performing causal analysis, we represent the association between variables across different basins as networks. We demonstrate the causal relationships between land variables across basins through land-atmosphere, atmosphere-atmosphere, and atmosphere-land interactions showing that the basins are not hydrologically independent. A perturbation in a river basin due to the proposed interlinking can travel to the neighboring basins by atmospheric pathways. Further, we used a modified regional climate model–Weather Research and Forecast coupled with Community Land Model 4 (WRF-CLM4, details in "Methods")[41]–to test the hypothesis that by land-atmosphere feedback, the additional irrigation from river interlinking can lead to changes in the Indian summer monsoon spatial patterns and the hydrology of the neighboring basins. To our knowledge, such feedbacks have not been considered in the literature for any globally existing or planned interlinking projects.

This study shows that river basins are linked to each other by land-atmosphere feedback, and any perturbation in one basin can travel to neighboring basins. This result is at odds with the conventional assumption of the absence of atmospheric links between river basins while planning hydrological projects. We also show that by land-atmosphere feedback, river interlinking projects in India will affect the Indian summer monsoon leading to a reduction in September rainfall in dry regions of the country, further aggravating the water stress. The methodology and results presented here pave the way for similar scientific impact assessments of river interlinking and other large-scale hydrological projects across the globe.

## Results and discussion
### Information links between the basins
Figure 1c shows the causal links between land variables across basins found using PCMCI (see "Methods") on 40 years of continuous daily reanalysis data from ERA-5 (1981–2020). Links are shown only if found statistically significant at a 99% confidence level. Each arc is a variable, and the arrow represents the link's direction to the other variable it projects onto. We get many links between land variables across all basins, indicating that surface soil moisture is connected across different river basins. Figure 1c shows that some basins show more outgoing links than incoming links. For example, latent heat flux from the Ganga basin, LH_G, has outgoing links to almost all river basins. However, it has incoming links only from land variables of the Mahanadi and Godavari basins. High recycled precipitation due to land-atmosphere feedback is well established for the Ganga basin[32,33,42]. Cauvery basin, on the other hand, has a large number of incoming links from all other basins, as evident from Fig. 1c. Literature shows that the Cauvery basin receives recycled precipitation generated by evapotranspiration from the neighboring regions[43]. Stronger causal connections exist between the land variables of other basins (Figs. 1c, S2). This means that the river basins can have characteristic properties of

being 'donor basins' or 'recipient basins' depending on the net transfer of moisture through atmospheric pathways. Supplementary table ST3 shows the sign of each link in Fig. 1c. We observe both positive and negative land-atmosphere feedback across different river basins. For example, land variables from Ganga have positive links to land variables of the Mahanadi basin but negative links to land variables of the Cauvery basin. The presence of both positive and negative connections between different river basins indicates that soil moisture from one basin may reduce or amplify soil moisture in another basin. We also generated the network between land variables using Granger Causality (GC) and Transfer Entropy (TE), and the results are shown in Fig. S2 (a, b, respectively). GC shows the highest number of links with all variables connected at a 99% confidence level (Fig. S2a). TE is argued to be a non-linear extension of GC[44]. We observe fewer links using TE (Fig. S2b) compared to GC because, while conditioning, it can remove strong non-linear autocorrelations. Since both methods are bivariate, they fail to resolve for cross-correlation effects of other variables in a high dimensional dataset; PCMCI is known to be superior to these methods (even their multivariate extensions)[45]. However, both TE and PCMCI generate almost the same number of links (Figs. 1c and S2b), which indicates the presence of common drivers or indirect links outside the land variables.

The above-discussed analysis shows causal relationships across land variables. Since land variables cannot directly transfer water from one basin to another, the links we see in Fig. 1c, and S2, are indirect links or are showing due to common drivers such as the El Niño-Southern Oscillation (ENSO). ENSO is known to control ISM and hence can simultaneously control soil moisture in different basins. A causal delineation technique would show a connection between basins unless the confounder (here ENSO) is included in the conditioning set. Actual links might be from ENSO (or Precipitation) to soil moisture individually for the three basins, which we fail to capture. It is also possible that the local changes in soil moisture in a river basin may indirectly affect that in another river basin through land-atmosphere feedback (called an indirect link because the true link from land-atmosphere feedback is getting delineated as links between land variables as seen in Fig. 1c), which in turn would affect the large-scale flow. The presence of indirect links (or confounders) is confirmed by Fig. 1c, generated using PCMCI. Since PCMCI conditions on all necessary variables from the input set ("Methods"), it should show fewer links than Granger Causality (GC, Fig. S2a) or Transfer Entropy (TE, Fig. S2b), if confounding variables were present among land variables, which is not the case. Hence, from the links in Fig. 1c, it can be concluded that the variables forming the pathways between land variables need to be included to find true causal linkages between different river basins.

To address these challenges, we include atmospheric variables and generate a causal network using a non-linear estimator of PCMCI (see "Methods" section for details), which tries to account for common drivers while controlling the high dimensionality. The monsoon drives the climate and the water cycle in India. To understand the land-atmosphere processes and their impacts on the water cycle, we have performed an analysis using PCMCI for the summer monsoon season. We have applied PCMCI to all the land and atmospheric variables from reanalysis data separately for each year's monsoon seasons with 122 days each, considering a maximum of 10 days' lag. Figure 2 presents the links which were found statistically significant at 95% confidence ($p < 0.05$) for more than 20 out of 40 years (1981–2020). We hypothesized that the causal connections between land variables of two different basins, A and B, exist through a series of indirect links: land variable (river basin A) → atmospheric variable (river basin A) → atmospheric variable (river basin B) → land variable (river basin B). We present the links in the same way in Fig. 2. The nodes in the leftmost column are the source land variables. The links from these nodes go to the second layers, the atmospheric variables of the same basin. For example, the link originating from the latent heat of Ganga (LH_G) goes

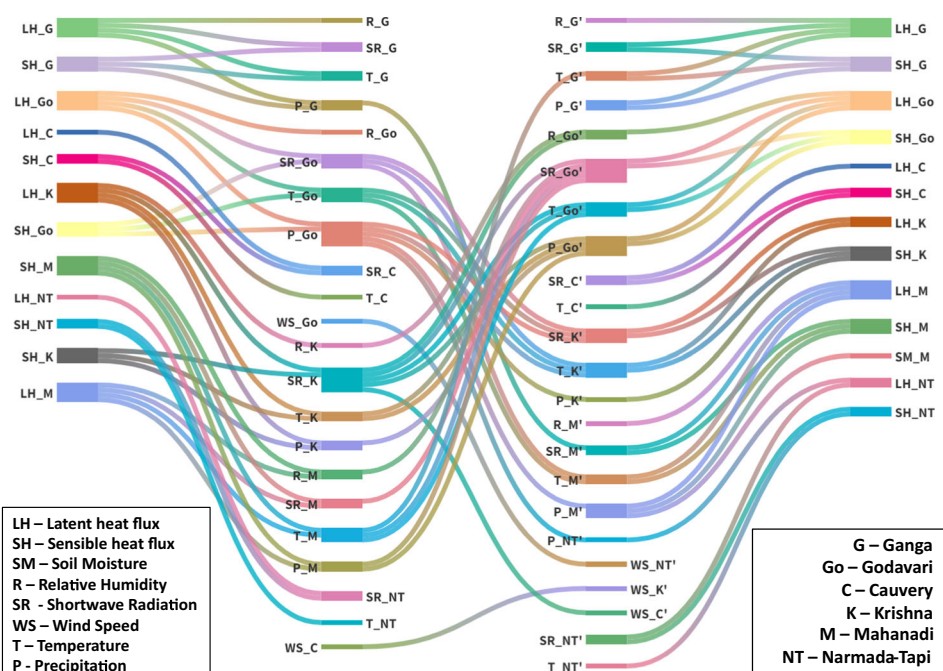

**Fig. 2 | Inter-basin connections via land-atmosphere feedback.** Connections using PCMCI from Land to atmosphere within basin (first column), between atmospheric variables across all basins (second column), and from atmosphere to land within the basin (third column). A link is shown only if it is found statistically significant at 5% level more than 50% of the time (20 years out of 40 years (1981–2020)). Names are variable symbols followed by the basin they belong to for example, LH_G means latent heat flux from Ganga basin. The first and second column of variables are land variables (soil moisture SM, latent heat flux LH, and sensible heat flux SH) and atmosphere variables (precipitation P, temperature T, relative humidity R, wind speed WS, and incoming short wave radiation SR), respectively, links between which represent land-to-atmosphere connections within each basin. Links between next two columns represent atmosphere to atmosphere connections, for example, there is a link from temperature in Ganga basin (T_Go) to that of Mahanadi basin (T_M). Links in the last column represent downward connections from atmospheric variables to land variables within basin.

to the temperature, precipitation, and humidity of Ganga (T_G, P_G, R_G), showing that the evapotranspiration (ET) of Ganga basins contributes to the moisture content in the air and hence, temperature and precipitation received within the Basin. This precipitation is called recycled precipitation[32]. We observe direct within-basin links from LH to P in Godavari, Krishna, and Mahanadi, indicating that these basins might have a high within-basin recycling ratio (ratio of precipitation caused by within-basin ET to the total precipitation in that basin). We also see that in some basins (like Cauvery and Narmada-Tapi), there is no link from LH to P; however, there are links from LH to R and T. This might indicate that ET leads to changes in the moisture content of the air however, it doesn't always translate into recycled precipitation. In such cases, the moisture supplied to air is transported to other basins, which should reflect as links between atmospheric variables of different basins (Fig. 2, 2nd column to 3rd column). This result is consistent with PCMCI showing some river basins with higher outgoing links than others in Fig. 1c. The land variables cannot cross the river basin boundary, but the atmospheric variables can. The 3rd vertical layer from the left presents the atmospheric variables of the information-receiving basins. The "apostrophe" sign after a basin symbol (for example, M for Mahanadi in T_M') signifies the receiving basin's atmospheric variable. For example, there is a link from precipitation in Ganga (P_G) basin to incoming radiation in the Mahanadi basin (SR_M'), which indicates that moisture from the Ganga basin might contribute to cloud formation in the Mahanadi basin. Precipitation from the Godavari basin (P_Go) is connected to precipitation from the Krishna, Mahanadi, and Narmada-Tapi basin (P_K', P_M', P_NT'). The 4th layer, or the rightmost layer, contains the land variable of the receiving basin. The temperature of the Mahanadi basin (T_M') impacts the latent and sensible heats of the Mahanadi basin (LH_M, SH_M). Hence the pathway from the Ganga basin to the Mahanadi basin (as seen as a link from LH_G to LH_M in Fig. 1) can be traced as LH_G→P_G→SR_M'→LH_M.

Interestingly, we also see a similar pathway from the Mahanadi to that of the Ganga, indicating feedback.

To make sure that the links are not just artifacts of the sample and are actual characteristics of the system, we applied PCMCI to another reanalysis data: Modern-Era Retrospective analysis for Research and Applications, Version 2 (MERRA-2)[46] and outputs of control runs (CTL) of regional climate model (WRF-CLM4, see "Methods"). The results are shown in Supplementary Figs. S3 and S4, respectively. We found that links in the networks derived from MERRA-2 and control runs of WRF have a clear resemblance to Fig. 2, though they are not the same. We also don't expect the networks from reanalysis data and the model outputs to be the same as the model runs contain irrigation which influences land-atmosphere feedback, whereas the reanalysis data do not. However, similarity in the causal pathways in the reanalysis dataset and the model runs indicates that land-atmosphere feedback leads to linkages between the river basins regardless of irrigation representation, which provides pathways for inter-basin water transfer to alter the ISM.

Figure 3 shows a simplified schematic explaining the mechanism of perturbation brought by river interlinking. The intra-basin land-to-atmosphere connection happens in the form of SM contributing to the moisture content of the air through evapotranspiration (high evapotranspiration during high SM) while also causing surface cooling (Fig. 3). The LH and SH control the atmosphere's moisture content (through wind, driven by temperature). The supplied moisture by evapotranspiration can lead to recycled precipitation in the same basin or can get transported to faraway regions by the wind. Evaporative cooling changes the thermal contrast between ocean and land or in between different land regions changing wind patterns and, subsequently, the moisture transport and rainfall. Earlier studies[47] suggested that surface cooling (due to other reasons like aerosols) leads to a decline in monsoon rainfall in India. The increased recycled

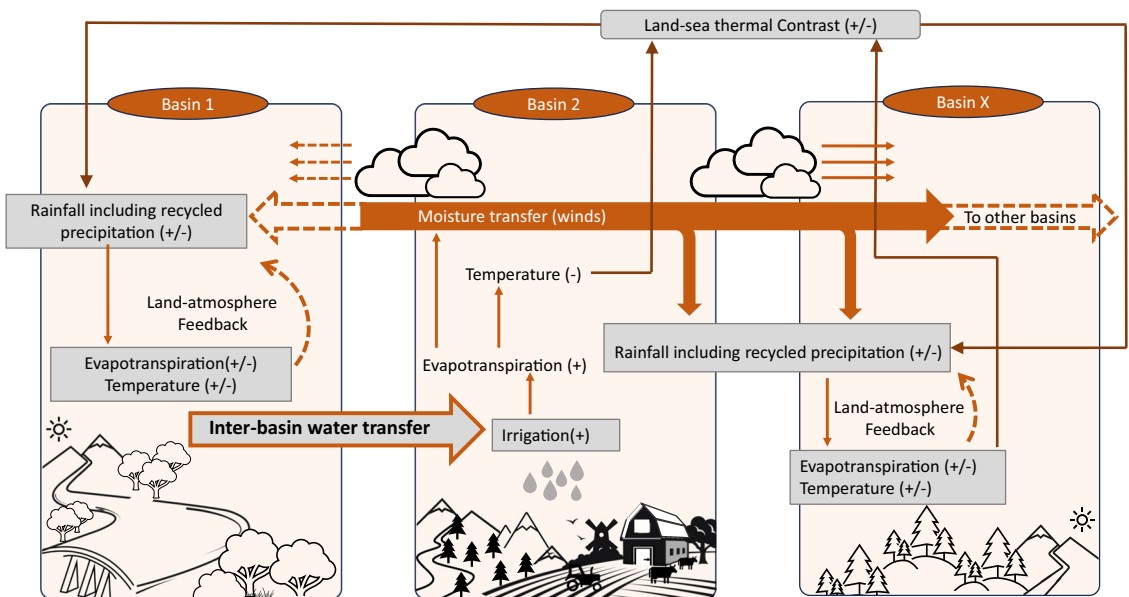

**Fig. 3 | Schematic diagram explaining the land-atmosphere feedback and changes in monsoon rainfall in response to river-interlinking.** The perturbations in the land water management leading from the inter-basin water transfer impact the spatial pattern of rainfall on the distant basins. The intra-basin land-to-atmosphere connection happens in the form of soil moisture contributing to the moisture content of the air through evapotranspiration (high evapotranspiration during high soil moisture (SM)) while also causing surface cooling. The supplied moisture by evapotranspiration can lead to recycled precipitation in the same basin or can get transported to faraway regions by the wind, which can then change the precipitation patterns of the region. Evaporative cooling changes the thermal contrast between ocean and land or in between different land regions changing wind patterns and, subsequently, the moisture transport and rainfall.

precipitation may have the opposite impact of increasing precipitation, and their resultant processes are quite complex. Overall, the perturbations in the land water management leading from the inter-basin water transfer impacts the spatial pattern of rainfall on the land (Fig. 3). Hence, for upward links (first column in Figs. 2, S3, and S4), in most cases, PCMCI detects LH as the source variable instead of SM. Since SM can be directly impacted by precipitation, we can expect direct downward links from the atmosphere to SM (Figs. 2, S3, and S4). However, this signal is weak because it is short-lived and dependent on the duration of precipitation spells. Atmospheric variables possess long and sustained controls on soil moisture by modulating latent heat fluxes through winds, clouds, etc.; hence, in the case of downward land-atmosphere connections, links are primarily toward heat fluxes (Figs. 2, S3, and S4). The atmosphere-to-atmosphere interactions between different basins occur through moisture and heat transported by winds across basin boundaries (Fig. 3).

## Feedback from the proposed interlinking

Based on the above causal analysis, we hypothesize that a perturbation in the land variables of a receiver basin due to the proposed interlinking can also affect its neighboring basins (for example, feedback between Ganga and Mahanadi explained above), including the donor basin through land-atmosphere feedbacks, which altogether, can lead to changes in spatial patterns of Indian Summer Monsoon (ISM). To test the hypothesis with a physics-based model, we used the coupled land-atmosphere model WRF-CLM4 (Details in "Methods"). We have chosen the period of the Indian Summer Monsoon (15 May to 31 October, initial 16 days from 15 May to 31 May used as spin-up run every year) from 1991 to 2012 to see the potential impacts of surplus irrigation by interlinking projects on other basins (see "Methods"). Our control run (hereafter CTL) contains the currently practiced irrigation in India as obtained from the agricultural census data and has previously been demonstrated to possess a reasonable skill in simulating Indian summer monsoon[48]. We performed another simulation (the irrigation run, hereafter IRR) by increasing the percentage of irrigated area to 80% in regions where interlinking projects target an increase in

the culturable command area, as shown in Fig. 1b. Figure 1b shows the increase in the percentage of irrigated area in each grid cell to achieve 80% irrigated area in the IRR run. The IRR run provides irrigation (in addition to irrigation in the CTL run) of 600 mm (around 4 mm per day) for normal crops and 1450 mm (12 mm per day) for paddy to an area of about 30 million hectares across the country (Fig. 1b). The simulations consider the India-specific crop and irrigation practices[48–50] (details in "Methods"). The differences in results between the two simulations provide conclusive evidence of the feedback from the interlinking to ISM through land-atmosphere interactions.

Figure S5 shows the difference in mean daily precipitation between IRR and CTL runs (IRR-CTL) for the monsoon season (b); June to September, JJAS), June (c), July (d), August (e), September (f). Hatched lines in plots indicate the regions where the difference is statistically significant at a 90% confidence level. We observe small regions of statistically significant increase and decrease in precipitation during JJA (Fig. S5c–e). During July (Fig. S5d), we find an increase in rainfall over the eastern parts of the Ganga basin. At the same time, there is also a substantial decline in precipitation in the Narmada-Tapi basin and western parts of the Ganga basin. Overall, there is no spatially consistent increase/decrease in precipitation during JJA. September sees the most widespread and maximum statistically significant reduction in precipitation (Fig. S5b). This is also evident from mean monsoon rainfall (Fig. S5b), which shows a statistically significant change in precipitation that is spatially similar to September month (Fig. S5f). JJA rainfall changes cannot compensate for the decrease in September rainfall as we don't observe any spatially consistent widespread statistically significant increase in JJA (Fig. S5). The simulated changes in September rainfall can be attributed to land-atmosphere feedback. The mean daily precipitation during September shows a statistically significant reduction of up to 4 mm/day in central India and some parts of north and western India (S5, f). The contribution of the land-atmosphere feedback to the Indian monsoon is maximum in September due to the widespread high soil moisture (resulting from JJA rain) and matured crop conditions (Kharif season crop) in September[32,51]. This is also consistent with our results which show

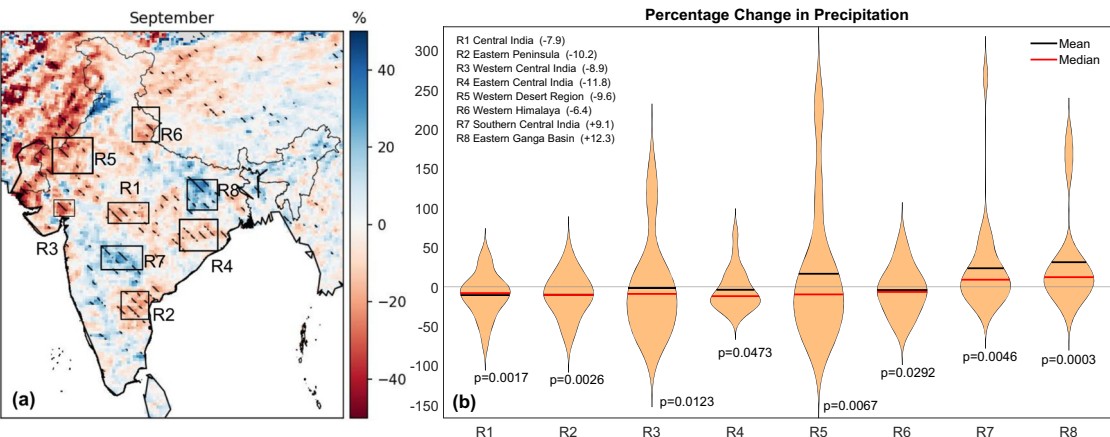

**Fig. 4 | Change in September precipitation after river-interlinking. a** Percentage change in mean daily precipitation between WRF irrigation run (IRR) and control run (CTL) (IRR-CTL) for the month of September. Hatch lines mark regions where the difference was found statistically significant at 90% confidence tested on 660 data points. **b** Violin plots of percentage change for all years in mean September rainfall for regions marked in (**a**). Median and mean change for all years are represented by red and black horizontal lines, respectively. Median percentage change is also written with the name of each region. There is a significant reduction in September precipitation of up to 12% in central (state of Madhya Pradesh), eastern (states of Odisha and Chhattisgarh), northern (state of Uttarakhand in western Himalaya), and western arid region (states of Rajasthan and Gujarat) of India.

minimal statistically significant changes in precipitation during JJA (Fig. S5). After September, the long dry period starts in the country with another cropping season, the Rabi season, in November–December. Hence, the rainfall and subsequent recharge in September has a longer-term impact on meeting the water demand in the non-monsoon months and reducing water scarcity. Hence, we have selected September month for further analysis.

Figure 4a shows the spatial distribution of mean percentage change in September rainfall for various regions in India for all years. The statistical significance of mean precipitation change was tested using a t-test on September precipitation from CTL and IRR runs (660 data points). Statistically significant regions have been hatched in the figure. The regions which experience statistically significant ($p < 0.05$) declines in September monsoon rainfall are central India (R1), east peninsular India (R2), coastal Gujrat (R3), east-central India (R4), the dry western region of Rajasthan (R5), western Himalayan foothills in Uttarakhand (R6). Figure 4b shows the probability distribution of percentage change in these regions as violin plots with the median and mean value shown by a red and black colored horizontal line, respectively, and $p$-values of the t-test on mean precipitation change written below each violin plot. Notably, the distribution of precipitation changes in these regions contains huge tails on the positive changes, particularly R3, R4, R5, R7, and R8. They experience sporadic events of precipitation increase in the form of extreme precipitation (more than 100% increase) with a systematic precipitation decrease where most of the values in the distribution are negative. Due to this, while a region experiences multiple years of precipitation deficits, we underestimate the precipitation reduction due to the heavy tails of surplus years. For example, in R3, R4, and R5, the precipitation reduction with the highest frequency is between 20–30%, whereas the mean value in these regions reports negligible or a positive change which could be misleading. Hence, while the driest region of India, the western region of Rajasthan (Fig. 4, R4), shows a mean increase in September precipitation of around 10%, it experiences a 10% median decline in rainfall. The highest median reduction is about 12 % in region R4 (the state of Odisha), followed by 10% in R2 (the state of Andhra Pradesh), 10% in R5 (the state of Rajasthan), and 9% in R3 (the state of Gujarat). The rainfall in central India, a part of the core monsoon zone, also shows an 8% decline in the simulated September rainfall due to interlinking. The western Himalayan foothills in Uttarakhand and east-central India also shows a moderate decline (6.4%) in September rainfall due to excess irrigation from the proposed interlinking. It is

worth noting that while there is a reduction in September precipitation in generally dry parts of the country, there is also an increase in September precipitation by up to 12% in east India (R8, states of Bihar, Jharkhand, and eastern Uttar Pradesh) and up to 10% in parts of the Deccan plateau (R7, states of Maharashtra and Telangana). Hence, there is conclusive evidence that river interlinking projects can alter the spatial pattern of ISM rainfall.

Moreover, the interlinking will also result in a changing spatial pattern of temperature over India (Fig. S6, a). Figure S6 shows the difference in mean daily values for daily maximum temperature, surface latent heat flux(b), and root zone soil moisture (c) between IRR and CTL runs for September. The changing meteorological patterns result in statistically significant mean monthly soil moisture and latent heat flux changes. However, there is a lack of one-to-one consistency everywhere due to complex hydrometeorological processes. The regions with less precipitation are accompanied by an increase in daily maximum temperatures of up to 1 °C. and a decrease in soil moisture of around 15 mm (Fig. S6; a, c, respectively). The changes in soil moisture in the grids receiving daily irrigation cannot be used to quantify the impacts of land-atmospheric feedback and hence, masked with gray color. The irrigated grids are also visible as having high latent heat flux in Fig. S6b. The proposed interlinking aims to improve soil moisture everywhere by taking surplus runoff from surplus regions and irrigating the deficit regions. However, contrary to this expectation, the feedback from the extra irrigation at the deficit basins results in declining rainfall in many neighboring areas, with a decline in soil moisture there. While there is a post-interlinking increase in soil moisture in the Krishna basin, the western portion of the Godavari and Narmada-Tapi basins, and the eastern Ganga basin, there is a pronounced decline in soil moisture of the Mahanadi, Godavari, and western part of Ganga basin in Indian states of Odisha, Chhattisgarh, norther Maharashtra, Madhya Pradesh, and Rajasthan. Hence, the purpose of improving soil moisture in deficit regions is met but at the cost of declining available water in the neighboring regions. Such unexpected feedbacks were unforeseen in the planning stage, and we contend that it is necessary to consider the land-atmosphere feedback processes in arriving at the policy decisions related to the interlinking.

El Niño Southern Oscillation (ENSO) is a significant driver of interannual variability of the monsoon rainfall over India. To understand the interannual variations of the land-atmosphere feedback from the excess irrigation proposed through interlinking, we separately analyzed El Niño and La Niña years (ST2, Fig. S7 for El Niño and Fig. S8

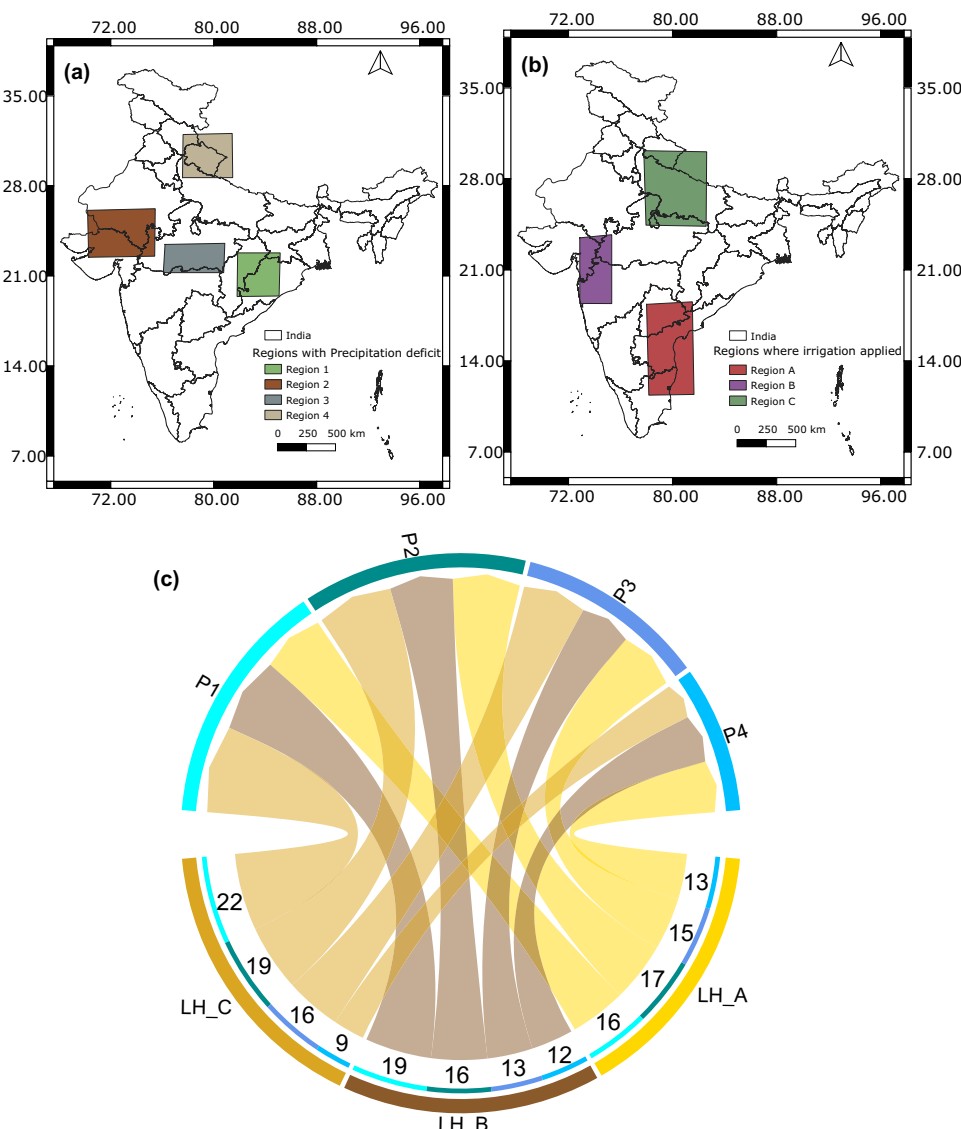

**Fig. 5 | Causal connections from river-interlinking to reduced precipitation.**
a Selected regions where irrigation was applied in irrigation run (IRR). **b** Selected regions of September precipitation where drying is observed due to interlinking. **c** Connections form change in latent heat flux (IRR - control run (CTL)) in the irrigated regions to change in precipitation (IRR-CTL) in the highlighted regions

using Transfer Entropy. Links are labeled as the number of years, when they were found out to be statistically significant ($p < 0.05$) out of 22 years of simulations (1991–2012). This shows that change in precipitation from CTL to IRR experiment is causally related to the corresponding change in latent heat flux of other regions indicating consistent land-atmosphere feedback.

for La Niña). These include both canonical and Modoki types of ENSOs, given that the impacts are qualitatively the same[52]. We have produced the differences in mean daily values of precipitation (a), daily maximum temperature (b), root zone soil moisture (c), and surface latent heat flux (d) between IRR and CTL runs in both the supplementary figures (Figs. S7, S8) for September. Interestingly, soil moisture drying due to excess irrigation is more prominent in La Niña years compared to the El Niño years. The whole central Indian belt from the desert regions of Rajasthan to the eastern coast shows a decline in rainfall and temperature increase; hence, soil moisture declines in the La Niña years (Fig. S8). The dry western region shows a decline in rainfall and soil moisture even for the El Niño years with an increase in temperature (Fig. S7). The drying of the arid region due to interlinking could be alarming and hence, needs to be addressed in the planning for interlinking. Central Indian regions show an improvement in the rain due to interlinking in El Niño, which is good for the dry years. Overall, we found that the perturbed water management from the proposed interlinking can lead to changes in the spatial distribution of the Indian

Summer monsoon and a systematic reduction of precipitation in many regions, including the dry arid regions.

Further, to demonstrate the causal relationship between river-interlinking and the Indian summer monsoon, we find causal connections from the regions with surplus irrigation to regions that experience drying in the IRR experiments. Figure 5a shows three chosen regions (southern peninsula—region A, western India—region B, and a part of Ganga basin—region C) out of areas where irrigation was applied, and Fig. 5b shows a few selected regions where maximum drying was witnessed (Region 1—central-eastern India, Region 2—Central India, Region 3—western India, and Region 4—western Himalayas). Both variables, i.e., LH from regions A, B, and C and P from regions 1, 2, 3, and 4, were taken as differences between CTL and IRR runs. The results are shown in Fig. 5. We tried to find the causal links from the latent heat flux (LH, IRR-CTL) of regions A, B, and C to precipitation (P, IRR-CTL) from regions 1, 2, 3, and 4. We use TE here, as we want to capture both direct and indirect connections in model simulations. The link thickness and labels written at the beginning of the

link show the number of times the link was found statistically significant ($p < 0.05$) out of 22 years of simulations. The node names are written as the variable symbol followed by the region they belong to. For example, P1 means precipitation over region one, and LH_A means latent heat flux in region A. The presence of causal links from LH to P from all regions indicates the influence of irrigation at regions A, B, and C on precipitation over regions 1, 2, 3, and 4 via land-atmosphere feedback. It shows that the reduction in precipitation from CTL to IRR experiment is causally linked to extra irrigation applied in the IRR experiment. The most robust links (present during most number of years) from all three irrigated regions are toward P1. The link from LH_C to P1 has the highest consistency and was found across all years. Our irrigation feedback results are consistent with the earlier studies that show the Ganga basin is the global land-atmosphere feedback hotspot[32,42,51,53]. Figure S9 shows similar results for El Niño (Fig. S9, a) and La Niña (Fig. S9, b) years. Considering a link as consistent if it was found statistically significant ($p < 0.05$) in more than 50% of years (at least 3 out of 5 years), the connection from the LH_A, LH_B, and LH_C toward P1, P4, and P2 remain consistent for both El Niño and La Nina years. While links to P3 are consistent during El Niño years, they were found to be less consistent from LH_B and LH_C during La Niña. The presence of a link here is a measure of the consistency of land-atmosphere feedback and cannot be attributed to the relative strength of drying during El Niño and La Niña years. Our results show that the land-atmosphere feedback from irrigated areas, especially from the Ganga basin and southern peninsula, remains consistent during El Niño and La Niña years.

## Implications of interlinking projects

India has a rapidly growing problem of water stress due to global warming, population growth, pollution, and change in land use. As per the Central Water Commission, Government of India, the current per capita availability of water in India is around 1400 cubic meters, slated to reduce to about 1200 cubic meters by 2050, and a large portion of the country is already classified as water stressed[54,55]. A large fraction of India's water resources is used for irrigation in agriculture. The water demand will further increase with the rapid intensification of agriculture. As water demand is rising rapidly, within the next 20 years, India might need most of its runoff to meet its urban and agricultural needs[56]. As a solution to this problem, India has planned river-interlinking projects to transfer water from surplus to deficit basins to cater to the water demand of the growing population. The goal is to keep the maximum possible water—which earlier used to reach oceans from river basins—on the land to meet the growing water demand of the country. The assumption behind such planning is that the river basins do not have atmospheric connections, and hence, feedback from linking a basin to other river basins will not affect the rainfall patterns in the source basin. Multiple studies[32,43,47,57,58] have demonstrated the local and distant impacts of land-atmosphere feedback. For example, one study[47] showed that aerosol forcing and subsequent land cooling is responsible for the reduction in precipitation over north central India. Excess irrigation also results in similar cooling, and together with increased recycling[32], it may have complex impacts on the Indian summer monsoon, which we study in the present work using regional modeling.

Here, we find that the assumptions made for the interlinking are not valid. The perturbed hydrological processes of the receiving river basins send feedback to the Indian monsoon, potentially changing the spatial patterns, specifically in September. Such changing patterns of monsoon, in turn, affect the hydrology of the neighboring basins. Hence, the hydrological processes across river basins are not independent, a critical result that most large-scale hydrological projects across the globe, including river-linking projects in India, do not consider while planning. The interdependence of river basins may significantly affect water demand-availability tradeoff within a basin. Our

WRF-CLM4 simulations attempt to quantify the possible changes in the Indian monsoon due to the proposed interlinking project. The results from these simulations show a systematic reduction of mean September precipitation of up to 12% in the western arid region (states of Rajasthan and Gujarat), central (state of Madhya Pradesh), central-eastern (states of Odisha and Chhattisgarh) and northern (states of Punjab, Haryana, and Uttarakhand) parts of India, which, based on our experimental setup, can be attributed to land-atmosphere feedback from interlinking. Given the minimal statistically significant change in precipitation in JJA and a reduction in September rainfall, precipitation in JJA cannot compensate for reduced water availability in September. The reduction in September precipitation will dry up the rivers in the subsequent months amplifying water stress manifolds in various parts of the country, which is an unexpected and unintended result of interlinking. Reduced September rainfall in a recipient basin in a water transfer scheme could increase its post-monsoon water demand more than anticipated at the design stage. Also, a reduction in precipitation of a donor basin can reduce its ability to send water to other deficit basins post-monsoon. Hence, we argue that the water balance of interlinked river basins needs to be carefully re-evaluated after including impacts of land-atmosphere feedback. The severity of water stress will depend on the socio-economy of the region where strong deficits are experienced. The majority of the population in the affected areas is dependent on agriculture. A reduction in monsoon rainfall can cause reduced crop yields in these regions increasing climate vulnerability and risk.

It is noteworthy that in the face of global warming, climate change, and rapidly changing land use and land cover (LULC), the reported inter-basin links will change in the future. Hence, this study does not intend to convey exact pathways between river basins; rather, we intend to demonstrate the role of land-atmosphere feedback from large-scale hydrological projects in perturbing large-scale processes like the Indian summer monsoon. In addition, we have not considered the feedback on the monsoon rainfall in response to the reduced runoff to the ocean due to interlinking. Recent studies show land-to-ocean runoff can perturb the monsoon rain[59] and may intensify the feedback quantified by us. In addition, the river basins can also exchange water via inter-catchment groundwater flows[60,61], incorporating which is outside the scope of the current study. Hence, proper quantification of the feedback from the proposed interlinking policy needs careful scientific investigation. This study is the first attempt to quantify the impacts of any large-scale hydrological project, like river interlinking on the Indian Summer Monsoon, which was not considered in planning these projects. Our results highlight the importance of regional land-atmosphere model-driven hypothesis testing and impact assessment while planning for large-scale hydrological projects.

## Methods
### Data

We use 40 years (1980–2019) of daily data from two sources (ST1): ERA-5 and MERRA-2 reanalysis products provided by the European Center for Medium-Range Weather Forecast (ECMWF, ERA-5)[38] and Modern-Era Retrospective analysis for Research and Applications Version 2 (MERRA-2)[46], respectively, over the major river basins of India (Fig. 1a), Ganga (G), Godavari (Go), Mahanadi (M), Krishna (K), Narmada-Tapi (NT—two river basins taken together), Cauvery (C). Since Narmada and Tapi are small basins, we club them together and represent them as a single basin. Variables considered are Latent Heat Flux (LH), Sensible Heat flux (SH), Precipitation (P), Relative Humidity (R), Wind Speed (WS) (resultant of u-wind (U), v-wind (V)), Incoming Shortwave Radiation (SR), Soil Moisture up to root zone (SM) and Temperature (T). Variables R, U, and V are taken at 850 hpa pressure level, and the remaining variables are near the surface. Variables were spatially averaged over each basin (Fig. 1a) and thus generated 48-time

series (8 variables on 6 basins) were converted to anomalies as deviations from their climatological mean values of 40 years. All variables were tested for the presence of non-stationarity using the Augmented Dickey-Fuller (ADF) test and were found to be stationary (statistically significant, $p < 0.01$). We use Oceanic Nino Index (ONI) values from National Oceanic and Atmospheric Administration (NOAA, https://origin.cpc.ncep.noaa.gov), USA, to classify El Niño and La Niña years as October to December ONI values of any year being above 1 and below −1, respectively (Supplementary Table ST2).

### Granger causality

Granger causality (GC)[34] is a causal delineation technique based on two conditions: first, that cause must precede the action, and second, cause must hold predictive information for the effect. A time series $X_t$ can be argued to 'granger cause' a time series $Y_t$ if past of $X_t$ can add unique predictive information for $Y_t$ when the predictive information from the past of $Y_t$ has already been accounted for. First, we regress the time series $Y_t$ with its past up to some lag τ followed by addition of past values of $X$ as regressors.

$$Y_t = c + \sum_{l=1}^{\tau} a_l Y_{t-l} + \varepsilon_t \qquad (1)$$

$$Y_t = c + \sum_{l=1}^{\tau} a_l Y_{t-l} + \sum_{l=1}^{\tau} b_l X_{t-l} + \eta_t \qquad (2)$$

If the second regression model is a better fit than the first one, it means that the past values of $X_t$ hold unique predictive information for $Y_t$ and $X_t$ can be argued to 'granger cause' $Y_t$. Above regression models are also called vector auto-regression (VAR) models. In this study, we perform pairwise GC analysis using VAR models of maximum lag 10 to test causal connections between land variables of different river basins. The best fit models were classified based on Akaike Information Criterion (AIC). We present a network using a binary adjacency matrix which contains value 1 at any $i$th row and $j$th column if $i$th variable granger found to granger cause $j$th variable at 99% statistical significance. The major assumptions of GC are that the data is stationary and can adequately be described using a linear model.

### Transfer entropy

Information exchange takes place between two variables ($X_t$ and $Y_t$ with time series $x_1, x_2, \ldots, x_t$ and $y_1, y_2, \ldots, y_t$), when a change in one variable leads to a change in another. This information exchange gets reflected as an overlap of Shannon's Entropy[62] of observed time series of those variables. Binning any time series $X_t$ into $m$ discrete parts, Shannon's Entropy can be computed as follows

$$H(X_t) = - \sum_{i=1}^{m} p_i(x_t)(\log(p_i(x_t))) \qquad (3)$$

Here $p_i(x_t)$ is probability of $x_t$ being in bin $i$. Transfer Entropy (TE) is an information-theoretic tool to delineate asymmetric connections in a non-linear dynamical system[63,64]. It is widely used in studying eco-hydrology and climate systems and is argued to be a well-suited measure of causality for such system[35,65–69]. It measures information from source variable to target variable while conditioning on the past of target variable, hence, it finds the dependence between two variables by excluding the effects from the history of the target variable. It can capture the non-linear and lagged causal connections between variables and can be considered a non-linear extension of Granger Causality[37]. Since TE measures overlap of Shannon's Entropy, it can be

estimated using $H(X_t)$ as follows

$$T(X_t > Y_t, \tau) = H(X_{t-\tau\Delta t}, Y_{t-\tau\Delta t}) + H(Y_t, Y_{t-\tau\Delta t}) - H(Y_{t-\tau\Delta t}) - H(X_{t-\tau\Delta t}, Y_t, Y_{t-\tau\Delta t}) \qquad (4)$$

Where, $T(X_t > Y_t, \tau)$ is Transfer Entropy from $X_t$ to $Y_t$ at lag τ, $H(X_{t-\tau\Delta t}, Y_t, Y_{t-\tau\Delta t})$ and $H(X_t, Y_t)$ are the joint entropies between variables computed using joint probabilities instead of marginals. TE can be normalized by entropy of a distribution with m number of bins where all bins are equally likely ($H_{max} = \log(m)$). This form of TE estimation has previously been used in literature[35,65,69,70]. In this study, number of bins taken is 11 which has been argued to be appropriate for measuring TE given sufficient data length[35], time step Δt is 1 (daily data) and lag τ varies from 1 to 10. After computing TE we test it for statistical significance at 99% confidence using method of shuffled surrogates[35,71–73] and present a network using a binary adjacency matrix which contains value 1 at any $i$th row and $j$th column if $i$th variable has statistically significant TE toward $j$th variable at 99% statistical significance.

Conditioning set in TE can include other variables in the system in addition to the past of target variable alone, however, with the increase in the number of variables, estimating multivariate TE becomes computationally expensive; hence, we stick to bivariate analysis. Assumptions while computing TE are that no other variables are influencing the target variables except source, the estimated probability density function (PDF) is close to the PDF of population, and the data is stationary.

### PCMCI

PCMCI algorithm belongs to a class of causal discovery methods called 'Causal Network Learning Algorithms' which first assume a fully connected causal graph and then iterate through each link testing for its removal by conditioning[37,45,74]. While TE faces trouble with high dimensional data, PCMCI handles the problem of high dimensionality[75] by dividing the process into two stages:

The first stage uses a modified PC algorithm (named after its inventors Peter and Clark)[76] to estimate the skeleton of the causal network. Given $\bar{X}_t = X_t^1, X_t^2, \ldots, X_t^N$, the set of all variables, to test for causality between all variable pairs from up to a maximum lag $\tau_{max}$, one needs to test for causality using maximum conditioning set of dimensions $N\tau_{max}$. This stage reduces the dimensionality of conditioning set by filtering out variables which have no significant contribution in the conditioning set. For each variable $\bar{X}_t^j \in \bar{X}_t$, after initializing preliminary parents $\bar{\mathcal{P}}(X_t^j) = (\bar{X}_{t-1}, \bar{X}_{t-2}, \ldots, \bar{X}_{t-\tau_{max}})$ the following hypothesis is tested for all variables $\bar{X}_{t-\tau}^i$ from $\bar{\mathcal{P}}(X_t^j)$:

$$\text{PC}: X_{t-\tau}^i \perp\!\!\!\perp X_t^j | \text{S} \qquad (5)$$

for any set S with cardinality $p$. Where S contains a subset of $\bar{\mathcal{P}}(X_t^j) \setminus \{X_{t-\tau}^i\}$. We keep on increasing $p$ and test the null hypothesis which if we fail to reject, the link is removed from $\mathcal{P}$. Hence, the first iteration ($p = 0$) removes uncorrelated variables from $\bar{\mathcal{P}}(X_t^j)$. In the second iteration ($p = 1$), variables which become independent after conditioning on the highest correlated variable from first iteration are removed from $\bar{\mathcal{P}}(X_t^j)$. In the third iteration ($p = 2$), those variables are removed from $\bar{\mathcal{P}}(X_t^j)$, which become independent after conditioning on two strongest drivers from the previous iteration and so on. A lenient alpha level of $\alpha = 0.2$ is taken for hypothesis testing in this stage so that true links are not lost. Thus, for each variable $X^j$, a reduced conditioning set is generated called 'Parents, $\bar{\mathcal{P}}(X_t^j)$', which contains all significant conditioning variables along with some false positives depending on our choice of statistical significance.

The second stage, called the momentary conditional independence (MCI) stage, finds causal connections for every pair $X_{t-\tau}^i \to X_t^j$

by conditioning on parents of $X_t^j$ and $X_{t-\tau}^i$ (generated in the first stage), for various time delays $\tau = \{1,2,\ldots,\tau_{max}\}$ and tests the following null hypothesis at $\alpha = 0.05$

$$\text{MCI}: X_{t-\tau}^i \perp\!\!\!\perp X_t^j | \bar{\mathcal{P}}(X_t^j) \setminus \{X_{t-\tau}^i\}, \bar{\mathcal{P}}(X_{t-\tau}^i) \forall X_{t-\tau}^i \in X_t^- \tag{6}$$

Where $X_t^- = (X_{t-1}, X_{t-1}, \ldots, X_{t-\tau_{max}})$, and $\bar{\mathcal{P}}(X_{t-\tau}^i)$ and $\bar{\mathcal{P}}(X_t^j)$ are the conditioning sets generated in the PC stage.

Both stages (PC and MCI stages) use conditional independence tests to measure the strength as well as statistical significance of connections. PCMCI can use a linear and non-linear test statistic based on partial correlation (ParCorr) and Conditional Mutual information using the k-nearest neighbor approach (CMI-knn)[77], respectively. In this study, we first find causal connections between land variables of different basins using TE, GC, and PCMCI on 40 years of continuous daily data (14,600 values). Here, we use ParCorr as our test statistic in PCMCI when testing for inter-basin land-land connections (Fig. 1c). We do not choose CMI-knn here as it involves estimation of multivariate probability density functions (PDFs) which becomes computationally expensive at high data lengths (18 variables of length 14,600 in this case). Next, we find land-atmosphere causal connections between different basins using PCMCI on land variables and atmospheric variables for each monsoon season separately (Fig. 2, ensemble of 40 samples containing 48 variables and 122 time-steps). In this step, we use CMI-knn as our choice of test statistic as we want to consider non-linear interactions as well. In all causal analysis performed in this study using various techniques maximum lag, $\tau_{max}$, is taken as 10 days. Each link goes through a two-stage robustness check. First, it is tested for statistical significance within PCMCI at a 95% confidence level. Next, once statistically significant links are found for each year, a link is considered robust and reliable if found more than 50% of the time. This two-stage robustness check strengthens the reliability of our judgment on a link between two river basins.

PCMCI is subject to a few assumptions[37,74] namely, stationarity of variables (time series are stationary), causal sufficiency (observed variables in the dataset are sufficient to capture the causal relationships among them), causal Markov condition (causal relationships among variables form a directed acyclic graph (DAG) and that there are no direct causal loops), presence of no contemporaneous causal links (no zero lag links), and faithfulness (no additional non-causal dependencies). In all causal delineation techniques, the fulfillment of causal sufficiency is left to the researcher's judgment. Earth science datasets become even more difficult to handle because systems are complex, with multiple interacting components and no boundaries. Primary processes governing land-atmosphere interactions are evapotranspiration (which supplies the moisture), advection (winds advect moisture to nearby/far-away places), and condensation (which then precipitates that moisture onto distant lands). Our study includes all the necessary variables reported in the literature that drive these processes as also documented in the literature[78]. Hence, we believe that taking three land variables and five atmospheric variables from each basin nears causal sufficiency for land-atmosphere interactions. Some external confounders like ENSO are not included because they bring about low frequency (interannual) variability in land-atmosphere processes, and since we performed the analysis for every year separately, including low-frequency external confounders is not needed. Hence, our dataset satisfies the primary assumptions of PCMCI to the best of our knowledge. Still, there is no denying that the data might violate other assumptions, for example, the assumption of independence of noise terms, which can also lead to spurious links. The results from causal networks reported in this study, hence, are subject to the validity of remaining above-mentioned assumptions as violation of an assumption can also lead to spurious links.

## WRF-CLM model setup

To simulate the surplus irrigation provided by river interlinking over the Indian region, we use a regional climate model - Weather Research and Forecasting model version-3 coupled with Community Land Model version 4 (WRF-CLM4)[41]. The choice of irrigation representation and amount of water applied are important considerations that affect land-atmosphere interactions[48,79]. Here we use a model set-up that represents realistic irrigation practices and water amounts to study the influence of inter-basin water transfer. We use a modified irrigation module in CLM4 that better represents the Indian practices of irrigation by incorporating ground withdrawal and flood irrigation practiced over paddy fields[48,80]. The domain used is from 59.5°E to 107°E and 3.7°S to 41.5°N (S3, f) with the model configured at 25 km grid spacing and 30 pressure levels in the vertical. We use initial and lateral boundary conditions from European Centre for Medium-Range Weather Forecast Interim Re-Analysis (ERA-Interim)[81] to perform two sets of experiments for 22 years (1991–2012) of the Indian summer monsoon (ISM; 15 May to 31 October). India receives almost 80% of its precipitation during monsoon and hence, any variation in ISM precipitation will translate into significant changes in water demand and supply from different basins which will directly impact inter-basin water transfer projects. Hence, we decided to study the impacts only during ISM. We perform monsoon simulation for each year separately as it is computationally efficient and has shown skill in simulating ISM with irrigation in previous studies[43,48,57,82]. Control experiment (CTL) prescribes irrigation water application over India using estimates from the agricultural census and a gridded reconstructed data[48,49]. The Irrigation experiment (IRR) adds additional water as irrigation by maximizing the irrigated area fractions on the grids which are going to benefit from interlinking. We incorporate surplus irrigation in the IRR run by converting crop plant functional types (PFTs) to irrigated crop PFTs in CLM4. We use MODIS vegetation classes in our simulations. Most of the cropland area in India corresponds to MODIS vegetation class 12, and 85% of the area of these grids are assigned to crop PFTs in CLM4. We increase the irrigated area in unirrigated PFTs to 80% of the grid area to generate a total irrigated area roughly equal to 30 million hectares, as proposed by the government of India. Figure 1b shows (highlighted) the approximate grids to receive the additional irrigation from the river interlinking project[23–25] with the increase in the percentage irrigated area from CTL to IRR run for each grid. Each grid cell projects an area of around 50,000 hectares and out of total 30 million hectares, around 20 million hectares belongs to regions fed by Himalayan rivers while around 10 million hectares belongs regions fed by peninsular rivers. The relative proportion of area irrigated from Himalayan and peninsular rivers is as per the publicly available interlinking plan and detailed project reports by the government of India[26]. The primary use of inter-basin water transfer is to increase the culturable command area across the country by around 30 million hectares, and here we consider the irrigation scenario in which the targets proposed in the DPRs are achieved. The amount of irrigation water added on these grids is estimated from the water availability and proposed culturable command area for the Kharif season from various projects[24] and is approximately equal to 600 mm (around 4 mm per day) for normal crops and 1450 mm (12 mm per day) for paddy. The model setup that we use has been evaluated for irrigation parameterization uncertainty in prior work[48]. In the present study, we utilize the same model set-up with realistic irrigation practices to apply extra irrigation to satisfy the targets of the planned inter-basin water transfer schemes.

A spin up time of 16 days (15 May to 31 May) is used in model runs to make sure the outputs are independent of initial conditions. To test the sensitivity of our simulations to the initializations, we performed simulations with varying spin-ups of 15, 30, 45, and 60 days for the year 2000, and the results are presented in Fig. S10. We find that the simulated precipitation for the monsoon period (June, July, August,

and September) is nearly the same in all four spin-up periods with minimal differences. Hence, a 15-day spin-up period, which has also been used in previous studies[48], is sufficient to generate stable estimates of precipitation during ISM. WRF simulations of the Indian monsoon are sensitive to the choice of convective parameterization scheme used. Here, we use the Betts-Janjic-Miller (BMJ) scheme which has been evaluated against other schemes and known to correlate well with spatial distribution of Indian Monsoon at daily scale. This scheme has also been reported to show reasonable skill in simulating the Indian Summer Monsoon in other studies[83,84]. To represent the sub-grid microphysics, planetary boundary layer, longwave, and shortwave radiation, we use Lin Scheme, Yonsei University Scheme (YSU), Rapid Radiative Transfer Model (RRTM), and Dudhia scheme, respectively. The parameterization is kept same for all simulations and the only difference between the two experiments (IRR and CTL) is the presence of additional irrigation in the IRR run to analyze changes in the simulated hydrometeorological variables during ISM.

## Data availability

ERA 5 reanalysis can be downloaded from https://cds.climate.copernicus.eu/cdsapp#!/dataset/reanalysis-era5-land. MERRA 2 reanalysis can be downloaded from https://gmao.gsfc.nasa.gov/reanalysis/MERRA-2/. Oceanic Nino Index (ONI) values are taken from https://origin.cpc.ncep.noaa.gov. The post-processed outputs from WRF-CLM4 CTL and IRR simulations generated in this study have been deposited in an open repository database under DOI 10.5281/zenodo.8246799[85]. The raw outputs from WRF-CLM4 simulations are available upon request. Source data for figures are provided with this manuscript. Source data are provided with this paper.

## Code availability

To perform PCMCI, a publicly available python package 'tigramite' (https://github.com/jakobrunge/tigramite) was used. Code from regional climate model WRF-CLM4 modified to incorporate India specific irrigation is available at GitHub https://github.com/IMMM-SFA/WRF_CLM4_Irrigation[80]. The co-authors of the present manuscript have prepared this India specific module. Sankey diagrams (Figs. 2, S3, and S4) have been plotted using a freely available online tool (https://flourish.studio).

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

## Acknowledgements

T.C. thanks Sachin Budakoti and Roshan Jha for their technical discussions on configuring the regional model. T.C. acknowledges the Prime Minister's Research Fellowship (PMRF), Ministry of Education, Government of India, for his PhD scholarship. K.A. acknowledges the Institute of Eminence-University of Hyderabad grant No.UOH/IOE: RC1-20-015. The work is financially supported by the Department of Science and Technology Swarnajayanti Fellowship Scheme, through project no. DST/SJF/E&ASA-01/2018-19; SB/SJF/2019-20/11, and Strategic Programs, Large Initiatives and Coordinated Action Enabler (SPLICE) and Climate Change Program through project no. DST/CCP/CoE/140/2018.

## Author contributions

S.G. conceived the idea and designed the problem. T.C. and S.G. designed the hypothesis testing method. T.C. and A.D. performed all the analyses. M.K.R. helped in the simulations. T.C. and S.G. primarily analyzed the results. K.A. and M.K.R. reviewed the results and provided suggestions. S.G. and T.C. wrote the manuscript. T.C., S.G., A.D., M.K.R., and K.A. reviewed the manuscript.

## Competing interests

The authors declare no competing interests.
