## [Peer Review File · Nature Communications]

REVIEWER COMMENTS

Reviewer #1 (Remarks to the Author):

The authors combined causal delineation techniques, a coupled regional climate model, and multiple reanalysis datasets to examine the land-atmosphere feedback of river interlinking projects on the Indian summer monsoon. Results show that land-atmosphere feedback generates causal pathways between river basins in India, and increases irrigation from the transferred water reduces mean rainfall in September by up to 12% in water-stressed regions of India. The topic is interesting and important for the impact assessment of large-scale hydrological projects. However, there are a number of significant limitations in the presented study.

Major points:

- (1) This study does not seem to have a theme throughout the text. The use of TE and PCMCI based on ERA5 data helps understand the causal pathways between river basins in India. However, there is no association between the causal pathways (illustrated by TE and PCMCI) and the modeling of land-atmosphere feedback on summer monsoon, i.e., the authors did not modify the mechanism entailed in the regional climate model, nor did they show whether the regional climate model has considered the demonstrated causal pathways. Furthermore, models used in ERA5 and MERRA2 do not incorporate the influence of irrigation; hence, the causal pathways between river basins presented based on these reanalysis data do not reflect the reality that occurred in India.
- (2) The rationality concerning the priority of transferred water for irrigation needs to be clarified. In river interlinking projects, there is no evidence to verify the use order of the transferred water. Meanwhile, it needs to be demonstrated why water transferred from surplus areas should be used to increase the proportion of irrigated areas up to 80% in water-scarce areas.
- (3) Changes in precipitation in a given month (i.e., September) of the year (even the decreasing ratio of 12%) do not seem to reflect the huge impact of the river interlinking projects. The enhanced soil moisture due to increased precipitation in the early phase of the monsoon season (June-August) may alleviate the water stress resulting from decreased precipitation in September. Therefore, the severity of the subsequent influence of decreases in precipitation in September should be clarified. Moreover, why not focus on precipitation changes throughout the summer monsoon period, that is, from June to September?
- (4) There are unreasonable designs in the methodology concerning the use of the coupled land-atmosphere model. The coupled land-atmosphere model (WRF-CLM4) was not running continuously,

but performed for each year separately. The effect of continuous operation of the river interlinking projects is not considered, so the simulation results may not be reliable. In addition, more clarification is needed as to why the spinning up for WRF-CLM4 was implemented for each year. The time (16 days from 15 May to 31 May) for spinning up of WRF-CLM4 is too short to achieve the equilibrium of the model.

Minor points:

Figure 2 is hard to understand and its content may need to be decomposed to highlight the main messages.

Reviewer #2 (Remarks to the Author):

This study comprehensively analyzes the changes in land-atmosphere feedback and the Indian summer monsoon due to massive river interlinking projects in India. The topic is quite important, the methods are fundamentally reasonable, and the findings are overall interesting. However, as for top journals such as NC, this manuscript still needs substantial revisions.

Major comments:

1. There is insufficient description and justification of the methods, especially for PCMCI part. The authors may provide more details about the methods themselves, as well as how they have been used to solve the problems in this study. Actually, I find that some sentences in the main text can be moved to the corresponding parts in Data and Methods.
2. More interpretation and contextualization of the findings should be provided. In Results and Discussion, the authors mainly present their new findings (as Results); however, few physical interpretations behind the findings (as Discussion) are provided. In particular, two of the used methods (i.e., TE and PCMCI) are both causal discovery methods with certain assumptions. It is usually easier to generate new findings based on these methods than to well explain the findings. Still, it is extremely important to provide rational analysis of the findings.

Specific comments:

1. Lines 111-113: "... hence, we stick to bivariate analysis." Some information may be missing by reducing the dimensions. Is it possible to analyze the effects of such a particular simplification on the results?
2. Lines 130-131: "Thus, the most likely reason for these links would be indirect links or common drivers." Any examples?

3. Figure 1c: for some variables in some river basins (e.g., SM_K, SM_G, SM_M, and SM_NT), many links are originated from them, but few links are pointed to them. Why? Would you please briefly explain this phenomenon?

4. Figure 2: some variables in the donor river basins (e.g., SM_Go, SM_C, and SM_NT) are missing (in the first column), while none of the variables is missing in the receiving river basins (in the fourth column). Any reasons for this? Moreover, I would suggest the authors re-order all the variables according to the basin names, like what they do in Figure S2.

5. Figure 4: why do the authors choose these regions? Are there any special features in these regions? I cannot find related sentences regarding this issue.

6. some typos:

Line 183: "aria" -> "area"

Line 371: "as" -> "has"

Line 383: "iterating" -> "iterates"

Line 411: "Indian monsoon season (ISM; ..."?

Reviewer #3 (Remarks to the Author):

Please see attachment.

Comments on
“River interlinking alters land-atmosphere feedback and changes the
Indian summer monsoon”

Manuscript # NCOMMS-22-46637
by Chauhan T. et al.

February 5, 2023

1 General Comments

The authors hypothesize in this study that water transfer between river basins may impact the donor or adjacent basins through land-atmosphere feedback. The authors claim that this hypothesis can be tested using causal delineation techniques, a coupled regional climate model, and multiple reanalysis datasets. The study is carried out in India. Moreover, authors also claim that their finding “disputes the conventional assumption of hydrological independence between river basins”.

The authors indicate that this study is a first attempt (I can't prove not disprove the statement at the moment) to test the hypothesis that large irrigation projects, under the ceteris-paribus-condition, may induce changes in neighboring river basins. To test this hypothesis, the authors use the WRF model under control experiments and analyse the data with two statistical approaches to test statistical causality (i.e., Transfer Entropy and Peter and Clark's Momentary Conditional Independence). Based on the analysis of results, the authors concluded that there is enough statistical evidence to reject the hypothesis that (1) “the hydrology of the river basins is independent” and that (2) “the feedback from interlinking river basins (through water project) will not affect the rainfall patterns in neighboring basins”.

The topic of this manuscript may be of interest for the readership of the Nat. Comm. Journal. At the moment, however there are many points that have to be clarified before publication.

2 Specific Comments

This paper has the following technical shortcomings:

- I do not agree with the statement that (L48ff) “the interlinking of the rivers assumes the river basins to be hydrologically independent”. It is in my opinion an overstatement. Hydrologically speaking, if two neighboring river basins do not have a common large scale aquifer (e.g. Karstic formation), we can assume the

that the generated superficial runoff is mostly dependent of the own contributing drainage areas. If there is intercatchment groundwater flows, the river basins are called non-conservative (NC). "There are many well-known physical reasons for the existence of NC catchments. Karstification of limestone is perhaps the most widespread reason worldwide" (Le Moine et al. WRR doi: 10.1029/2006WR005608, 2007). It is widely accepted in the hydrological community that we can't close the water balance only by measurements. Errors in measurements is one cause, the another one is leaking catchments. In south Germany and many basins in France this hypothesis not true (see also Zink et al. HESS 2017). Consequently, a generalization of basins independence is not a valid assumption.

In the present study, the authors have not checked or provide evidence that the basins used are conservative. This should be done.

- L184ff: 600 mm/day or 1400 mm/day is imposible. Authors probably mean per year. My doubt is confirmed in L424ff! Please correct.
- Please indicate why streamflow, one the key variables in water management, has not been included in this study? In my opinion, it should be included. Authors should also indicate the sign of the feedbacks (e.g. with color).
- I wonder why the authors excluded the well know approach proposed by Granger. They indicate that the TE and the PCMCI capture non-linearities, but there is a price for that: the increase of dimensionality and computational time. If the Granger Causality test is rejected, the authors could investigate non-linear causality pathways. I suggest the preset a map of the p-values of the Ho: X_t is not Granger causing Y_t . Instead of trying to find all potential connection, I would attempt to find the most obvious ones based on literature. Or to confirm the strong ones found by these two methods.
- I also don't understand why two methods are presented (TE and the PCMCI). What is the role of TE and PCMCI? Do both methods find the same causation pathways? Not clear for me. The authors should present the same kind of graph depicting the links. In Figure 4 I don't understand what the width of the link means? What is the statistical significance in Fig 4, and elsewhere?
- The authors use very sophisticated methods to investigate causal relationships but I can't read anywhere the p-values of the most important causal pathways. If there exist, p-values ≤ 0.05 for the links shown in Fig 2, why not to depict them?
- There are several publications in which it has been tested that irrigation "can affect" local climate (e.g., <https://gmd.copernicus.org/articles/13/3179/2020/gmd-13-3179-2020.pdf>). In this study, for example, the authors concluded that "Recent literature shows that irrigation mostly affects the near-surface variables, creating the so-called irrigation cooling effect.". It is advisable that the authors perform several sensitivity experiments with WFT and its several parameterization mechanisms to be able to conclude that these causation pathways are not simple artifacts of a small sample.

- The authors do not report confidence levels in any of the percentage increases (e.g., L209. This is a must in my opinion.
- I do not fully understand the experiment carried out by the authors. For example: 1) how many WRF simulations were carried out by the authors to account for parameter uncertainty and initialisation. 2) If a pathway is found in a given period of time, the authors should demonstrate that this pathway also occur in other locations or periods of time. 3) Authors should indicate how big is the sample size (ensemble of simulations or initializations).
- I suggest to present cross-validation experiments to demonstrate that a causation-pathways is not an artifact of a sample but a characteristic of the system. I don't see this at the moment.

3 Final Remarks

Based on the comments mentioned above and bearing in mind the publishing standards for a research article in a Nature Journal, I recommend **to return this paper to the Authors for major revisions.**

We thank the reviewers for their encouraging and constructive feedback. We have given a point-by-point response to all the reviewer's comments and made necessary changes to the manuscript in response to suggestions by reviewers. We are thankful to our reviewers and we believe that the comments provided have significantly improved the quality of our manuscript.

This response file provides complete documentation of the changes we made in response to each reviewer's comments. The document is designed so that the changes we have made in response to each comment can be immediately read and understood, independent of the other comments and responses. While this comprehensive comment-by-comment explanation requires some duplication of material throughout the document, we intend that it helps to swiftly and efficiently evaluate exactly how each comment has been addressed.

Reviewer comments are shown in bold. Author responses are shown in plain text.

Reviewer #1 (Remarks to the Author):

The authors combined causal delineation techniques, a coupled regional climate model, and multiple reanalysis datasets to examine the land-atmosphere feedback of river interlinking projects on the Indian summer monsoon. Results show that land-atmosphere feedback generates causal pathways between river basins in India, and increases irrigation from the transferred water reduces mean rainfall in September by up to 12% in water-stressed regions of India. The topic is interesting and important for the impact assessment of large-scale hydrological projects. However, there are a number of significant limitations in the presented study.

We thank the reviewer for the appreciation of the importance of our work, and useful comments provided. We have addressed all the limitations pointed out, and revised the manuscript accordingly.

Major points:

(1) This study does not seem to have a theme throughout the text. The use of TE and PCMCI based on ERA5 data helps understand the causal pathways

between river basins in India. However, there is no association between the causal pathways (illustrated by TE and PCMCI) and the modeling of land-atmosphere feedback on summer monsoon, i.e., the authors did not modify the mechanism entailed in the regional climate model, nor did they show whether the regional climate model has considered the demonstrated causal pathways. Furthermore, models used in ERA5 and MERRA2 do not incorporate the influence of irrigation; hence, the causal pathways between river basins presented based on these reanalysis data do not reflect the reality that occurred in India.

We thank the reviewer for this valuable comment. We acknowledge that analysing causal pathways and mechanisms in the model simulations is vital for establishing our premise, and we had not included this in the previous version of the manuscript. We would like to point out that the regional model considers irrigation, and agree that it is necessary to establish the land-to-atmosphere causal connections from the simulations using irrigation. It will also serve as a validation to ensure that the causal links are not just artifacts of a sample but a system characteristic. To address this concern, we applied the PCMCI technique to the outputs from our control runs (CTL) of the regional climate model (WRF-CLM4). The results are added in Supplementary Figure S4 of the manuscript and are also shown below. We show only the statistically significant links at $p = 0.05$, appearing at least 50% of the time (11 out of 22 years). Our CTL runs contain currently practiced irrigation scenarios in India. We found that links in the networks derived from ERA5, MERRA-2, and WRF CTL simulations resemble each other, though they are not the same. This solidifies our argument that land-atmospheric causal pathways connect river basins across their land boundaries. We have included this information in the results and discussion section of the manuscript (Lines 187- 198). Our particular thanks for this excellent suggestion.

We agree with the reviewer that the reanalysis datasets (ERA5 and MERRA2) do not contain irrigation. Indeed, due to the lack of irrigation in reanalysis datasets, the generated networks are expected to differ from our model simulations. This difference, in fact, provides a rationale behind, and the need for, conducting regional simulations with irrigation, and applying the causal discovery approach to the simulations. The

causal graphs obtained from the reanalysis data support the hypothesis that causal linkages exist across river basin land variables through land-atmosphere feedback regardless of irrigation. Hence, a change in land-water management in one basin due to inter-basin water transfer will also affect the other basins. We have updated the results and discussion section of the manuscript to clarify this (Lines 193-195).

The central idea of the manuscript is that land-atmosphere causal pathways exist between different basins which get perturbed by interlinking and might impact the Indian monsoon. We have modified the manuscript contents in the methods and results section to clarify this theme.

Supplementary Figure S4. Causal connections of land-atmosphere interactions across different basins in WRF-CTL runs using PCMCi (similar to Figure 2). A link is shown only if it is found statistically significant with $p < 0.05$ more than 50% of the time (11 years out of 22 years).

(2) The rationality concerning the priority of transferred water for irrigation needs to be clarified. In river interlinking projects, there is no evidence to verify the use order of the transferred water. Meanwhile, it needs to be demonstrated why water transferred from surplus areas should be used to increase the proportion of irrigated areas up to 80% in water-scarce areas.

We thank the reviewer for this comment and clarify the rationale behind the increased irrigated area in the model runs.

The detailed targets for all inter-basin water transfer projects in India are mentioned in the Detailed Project Reports (DPRs) and are publicly available at <http://www.nwda.gov.in/content/innerpage/detailed-project-report.php> (2022) (Reference 24). The reports explicitly mention that the primary goal of water transfer is to increase the culturable command area (CCA), which would be done by providing irrigation to previously unirrigated areas. The DPRs discuss each CCA's crop types and water use to set target water transfer requirements. Together, these projects target an increase of 30 million hectares in irrigated area, of which 20 million hectares belong to Himalayan rivers, and the rest belong to peninsular rivers. The extra irrigation provided in our simulation follows these proportions. While we agree with the reviewer that, in principle, the exact use of water transfer in a realistic situation may differ (in some cases, a large portion of water may be used for household purposes rather than irrigation), here we consider the scenario in which the irrigation targets proposed in the DPRs are achieved. We have mentioned this in lines 557 – 565 and (methods section) of the new manuscript.

Further, the rationale behind irrigating a grid up to 80% of its area is related to the conversion of MODIS vegetation classes to crop plant functional types (PFTs) in the Community Land Model-version4 (CLM4). Crop PFTs are assigned from two vegetation classes in CLM4 - croplands (class 12) and cropland/natural vegetation mosaic (class 14). Most of the cropland area in India corresponds to MODIS vegetation class 12, and 85% of the areas of these grids are assigned to crop PFTs in CLM4. We utilize grid cells where crop PFTs are not fully irrigated in the control run to add surplus irrigation post-water transfer. We increase the irrigated area to 80% on these grids to generate an extra irrigated area roughly equal to 30 million hectares, as

proposed by the government of India. Thus, we irrigate existing croplands in the model without converting other land use classes to irrigated croplands.

We have now included this information in the Data and Methods section of the manuscript (Lines 549- 555).

(3) Changes in precipitation in a given month (i.e., September) of the year (even the decreasing ratio of 12%) do not seem to reflect the huge impact of the river interlinking projects. The enhanced soil moisture due to increased precipitation in the early phase of the monsoon season (June-August) may alleviate the water stress resulting from decreased precipitation in September. Therefore, the severity of the subsequent influence of decreases in precipitation in September should be clarified. Moreover, why not focus on precipitation changes throughout the summer monsoon period, that is, from June to September?

We thank the reviewer for this comment and, in response, provide information about changes throughout the summer monsoon period and highlight the relevance of changes in September.

We agree with the reviewer that a surplus rainfall in JJA can alleviate the risk of deficits in September. However, there is no spatially consistent increase/decrease in precipitation during JJA. During July, for example, we find an increase in rainfall over parts of the Ganga basin (figure S5d). However, there is also a substantial decline in precipitation in some other parts of the Ganga basin and the Narmada-Tapi basin. The regions in the core monsoon zone (most of central India) do not experience much statistically significant change in JJA (figure S5 c-e) but show a significant reduction in September precipitation (figure 3 and S5 (f)). September sees a widespread and maximum statistically significant reduction in precipitation (figure S5 (b)). The imbalance between changes in JJA and September is also evident from mean monsoon rainfall (figure S5(b)), which shows a statistically significant change in precipitation, which is, importantly, spatially similar to September month (figure S5 (f)). Hence, the decrease in September rainfall cannot be compensated by JJA rainfall changes (supplementary figure S5). This information is now mentioned in the results and discussion sections of the manuscript (Lines 234 – 244, 386 – 388).

We agree with the reviewer that the relevance of September rainfall changes was not discussed in the earlier version. The contribution of the land-atmosphere feedback to the Indian monsoon is maximum in September due to the widespread high soil moisture (resulting from JJA rain) and matured crop conditions (Kharif season crop) (Pathak et al., 2017; Devanand et al., 2019; also added as references 32 and 47 in the revised manuscript). This is also consistent with our results which show minimal statistically significant changes in precipitation during JJA but a spatially consistent decline in precipitation during September (figure S5). After September, the long dry period starts in the country with another cropping season, the Rabi season, in November-December. Hence, the rainfall and subsequent recharge in September has a longer-term impact on meeting the water demand in the non-monsoon months and reducing water scarcity. Therefore, we focused on the feedback occurring in September. This has been mentioned in the manuscript results and discussion section (Lines 248 – 256, 391 – 397).

(4) There are unreasonable designs in the methodology concerning the use of the coupled land-atmosphere model. The coupled land-atmosphere model (WRF-CLM4) was not running continuously, but performed for each year separately. The effect of continuous operation of the river interlinking projects is not considered, so the simulation results may not be reliable. In addition, more clarification is needed as to why the spinning up for WRF-CLM4 was implemented for each year. The time (16 days from 15 May to 31 May) for spinning up of WRF-CLM4 is too short to achieve the equilibrium of the model.

We thank the reviewer for this suggestion and perform additional simulations to study the sensitivity to the spin-up period. The results confirm that the spin-up time we used is adequate.

To test the sensitivity of our simulations to the initializations, we performed WRF simulations with varying lengths of spin-up (15, 30, 45, and 60 days) for the year 2000, and the results (included as supplementary figure S10 in the revised manuscript) are shown below. The simulated precipitation for the monsoon period (June, July, August, and September) is nearly the same in all four spin-up periods, with minimal differences. Hence, a 15-day spin-up period, which has also been used in previous

studies (Devanand et al. 2019, reference 47 in the revised manuscript), is sufficient to generate stable estimates of precipitation during ISM. We have updated the methods section of the manuscript to include this information (lines 572 – 582).

We focus on the Indian summer monsoon (ISM) season in this study since India receives ~80% precipitation from the monsoon. Any variation in ISM precipitation will translate into significant changes in water demand and supply from different basins, directly impacting inter-basin water transfer projects. Understanding changes during the monsoon is of utmost importance to identify potential water-stressed areas in the future. Notably, most rivers in peninsular India are non-perineal and monsoon fed. Hence, we decide to study the impacts only during ISM. We perform separate runs of WRF-CLM4 for each year as it is computationally efficient. The credibility of this approach to simulate the Indian summer monsoon (ISM) with irrigation is well established in the literature (Paul et al. 2016, Paul et al. 2018, Devanand et al. 2019, Jha et al 2021; also present as references 43, 47, 77, and 78 in the revised manuscript). This is now mentioned in the methods section of the manuscript (lines 541 – 544).

S10: Precipitation from WRF CTL runs for the year 2000 with varying spin-ups of 15, 30, 45, and 60 days. A spin-up of 60 days means the WRF-CTL run was initialized from 1st April 2000.

Minor points:

Figure 2 is hard to understand and its content may need to be decomposed to highlight the main messages.

We thank the reviewer for this comment and, in response, simplify figure 2.

Figure 2 shows the land-atmosphere pathways within and across basins that establish the land-land connections observed in figure 1(c). The first layer of links at the left shows the within-basin links from land to atmosphere variables. The second layer of links in the middle demonstrates the cross-basin links among atmospheric variables. The 3rd layer of links in the right shows within the basin link from the atmosphere to land. To improve readability and to show only the most robust inter-basin links, we have now modified figure 2 to show links at a stricter threshold (only showing links that cross statistical significance at 5% level, appearing 50% of the times, i.e. $p < 0.05$ for 20 out of 40 years).

Figure 2: Connections using PCMC1 from Land to atmosphere within basin (first column), between atmospheric variables across all basins (second column), and from the atmosphere to land within the basin (third column). A link is shown only if it is found statistically significant at a 5% level more than 50% of the time (20 years out of 40 years (1981-2020)). Names are variable symbols followed by the basin they belong to, for example, LH_G means latent heat flux from the Ganga basin. The first and second columns of variables are land variables (soil moisture SM, latent heat flux LH, and sensible heat flux SH) and atmosphere variables (precipitation P, temperature T, relative humidity R, wind speed WS, and incoming short wave radiation SR) respectively, links between which represent land-to-atmosphere connections within each basin. Links between the next two columns represent atmosphere-to-atmosphere connections, for example, there is a link from the temperature in the Ganga basin (T_Go) to that of the Mahanadi basin (T_M'). Links in the last column represent downward connections from atmospheric variables to land variables within the basin.

Reviewer #2 (Remarks to the Author):

This study comprehensively analyses the changes in land-atmosphere feedback and the Indian summer monsoon due to massive river interlinking projects in India. The topic is quite important, the methods are fundamentally reasonable, and the findings are overall interesting. However, as for top journals such as NC, this manuscript still needs substantial revisions.

Major comments:

We thank the reviewer for his/her general appreciation of our work and its importance, and the useful comments. We have made substantial revisions to address all the points raised by the reviewer.

1. There is insufficient description and justification of the methods, especially for PCMCI part. The authors may provide more details about the methods themselves, as well as how they have been used to solve the problems in this study. Actually, I find that some sentences in the main text can be moved to the corresponding parts in Data and Methods.

We thank the reviewer for suggesting these improvements. In response, we have now improved the description of PCMCI in the methods section, and added discussion around Transfer Entropy (TE), Granger Causality (GC, now included as per the suggestion of a reviewer), and PCMCI along with their applicability in this study.

The description of PCMCI in the method section is more detailed now (Lines 487 – 516) and reads as follows:

1. "The first stage uses a modified PC algorithm^{72,73} to estimate the skeleton of the causal network. Given $\bar{X}_t = X_t^1, X_t^2, \dots, X_t^N$, the set of all variables, to test for causality between all variable pairs from up to a maximum lag τ_{max} , one needs to test for causality using maximum conditioning set of dimensions $N\tau_{max}$. This stage reduces the dimensionality of conditioning set by filtering out variables which have

no significant contribution in the conditioning set. For each variable $\bar{X}_t^j \in \bar{X}_t$, after initializing preliminary parents $\bar{\mathcal{P}}(X_t^j) = (\bar{X}_{t-1}, \bar{X}_{t-2}, \dots, \bar{X}_{t-\tau_{max}})$ the following hypothesis is tested for all variables $\bar{X}_{t-\tau}^i$ from $\bar{\mathcal{P}}(X_t^j)$:

$$PC: X_{t-\tau}^i \perp\!\!\!\perp X_t^j \mid S$$

for any set S with cardinality p . Where S contains a subset of $\bar{\mathcal{P}}(X_t^j) \setminus \{X_{t-\tau}^i\}$. We keep on increasing p and test the null hypothesis which if we fail to reject, the link is removed from \mathcal{P} . Hence, the first iteration ($p = 0$) removes uncorrelated variables from $\bar{\mathcal{P}}(X_t^j)$. In the second iteration ($p = 1$), variables which become independent after conditioning on the highest correlated variable from first iteration are removed from $\bar{\mathcal{P}}(X_t^j)$. In the third iteration ($p = 2$), those variables are removed from $\bar{\mathcal{P}}(X_t^j)$, which become independent after conditioning on two strongest drivers from the previous iteration and so on. A lenient alpha level of $\alpha = 0.2$ is taken for hypothesis testing in this stage so that true links are not lost. Thus, for each variable X^j , a reduced conditioning set is generated called 'Parents, $\bar{\mathcal{P}}(X_t^j)$ ', which contains all significant conditioning variables along with some false positives depending on our choice of statistical significance.

2. The second stage, called the momentary conditional independence (MCI) stage, finds causal connections for every pair $X_{t-\tau}^i \rightarrow X_t^j$ by conditioning on parents of X_t^j and $X_{t-\tau}^i$ (generated in the first stage), for various time delays $\tau = \{1, 2, \dots, \tau_{max}\}$ and tests the following null hypothesis at $\alpha = 0.05$

$$MCI: X_{t-\tau}^i \perp\!\!\!\perp X_t^j \mid \bar{\mathcal{P}}(X_t^j) \setminus \{X_{t-\tau}^i\}, \bar{\mathcal{P}}(X_{t-\tau}^i) \forall X_{t-\tau}^i \in X_t^-$$

Where $X_t^- = (X_{t-1}, X_{t-1}, \dots, X_{t-\tau_{max}})$ and $\bar{\mathcal{P}}(X_{t-\tau}^i)$, and $\bar{\mathcal{P}}(X_t^j)$ are the conditioning sets generated in the PC stage.

Both stages (PC and MCI) use conditional independence tests to measure the strength as well as statistical significance of causal connection. PCMCI contains an option to use a linear and non-linear test statistic called partial correlation (ParCorr) and Conditional Mutual information using the k-nearest neighbor approach (CMI-knn)⁷¹, respectively."

For description of TE in methods section, we have moved some lines from results and discussions (old submission) to methods sections (Lines: 459 – 462, new manuscript) and have increased the description of TE along with assumptions involved in computing TE (Lines 471 – 480).

We have updated the methods section (Lines 518 – 527) to give a complete description of steps followed in causal analysis which reads:

"In this study, we first find causal connections between land variables of different basins using TE, GC, and PCMCI on 40 years of continuous daily data (14600 values). Here, we use ParCorr as our test statistic in PCMCI when testing for inter-basin land-land connections (supplementary figure x). We do not choose CMI-knn here as it involves estimation of multivariate probability density functions (pdfs) which becomes computationally expensive at high data lengths (18 variables of length 14600 in this case). Next, we find land-atmosphere causal connections between different basins using PCMCI on land variables and atmospheric variables for each monsoon season separately (figure 2, ensemble of 40 samples containing 48 variables and 122 time-steps). In this step, we use CMI-knn as our choice of test statistic as we want to consider non-linear interactions as well. In all causal analysis performed in this study using various techniques maximum lag, τ_{max} , is taken as 10 days."

2. More interpretation and contextualization of the findings should be provided. In Results and Discussion, the authors mainly present their new findings (as Results); however, few physical interpretations behind the findings (as Discussion) are provided. In particular, two of the used methods (i.e., TE and PCMCI) are both causal discovery methods with certain assumptions. It is usually easier to generate new findings based on these methods than to well explain the findings. Still, it is extremely important to provide rational analysis of the findings.

We thank the reviewer for this useful comment. In response, we have now provided more discussion of the physical processes governing the links in figure 2.

We have expanded the discussion on the links generated using PCMCI in the results and discussion section, which now reads as follows:

" For example, the link originating from the latent heat of Ganga (LH_G) goes to the temperature, precipitation, and humidity of Ganga (T_G, P_G, R_G), showing that the evapotranspiration (ET) of Ganga basins contributes to the moisture content in the air and hence, temperature and precipitation received within the Basin. This precipitation is called recycled precipitation³². We observe direct within-basin links from LH to P in Godavari, Krishna, and Mahanadi, which indicates that these basins might have a high *within-basin recycling ratio* (ratio of precipitation caused by within-basin ET to the total precipitation in that basin). We also see that in some basins (like Cauvery and Narmada-Tapi), there is no link from LH to P; however, there are links from LH to R and T. This might indicate that ET leads to changes in the moisture content of the air however, it doesn't always translate into recycled precipitation. In such cases, the moisture supplied to air is transported to other basins, which should reflect as links between atmospheric variables of different basins (figure 2, 2nd column to 3rd column). This result is consistent with PCMC showing some river basins with higher outgoing links than others in figure 1(c). The land variables cannot cross the river basin boundary, but the atmospheric variables can. The 3rd vertical layer from the left presents the atmospheric variables of the information-receiving basins. The "apostrophe" sign after a basin symbol (for example, M for Mahanadi in T_M') signifies the receiving basin's atmospheric variable. For example, there is a link from precipitation in Ganga (P_G) basin to incoming radiation in the Mahanadi basin (SR_M'), which indicates that moisture from the Ganga basin might contribute to cloud formation in the Mahanadi basin. Precipitation from the Godavari basin (P_Go) is connected to precipitation of the Krishna, Mahanadi, and Narmada-Tapi basin (P_K', P_M', P_NT'). The 4th layer, or the rightmost layer, contains the land variable of the receiving basin. The temperature of the Mahanadi basin (T_M') impacts the latent and sensible heats of the Mahanadi basin (LH_M, SH_M). Hence the pathway from the Ganga basin to the Mahanadi basin (as seen as a link from LH_G to LH_M in figure 1) can be traced as LH_G→P_G→SR_M'→LH_M. " (Lines 161 – 185).

" From figure 1(c) it is clear that some basins show more outgoing links than incoming links. For example, latent heat flux from Ganga basin, LH_G, has outgoing links to almost all river basins, however, it has incoming links only from land variables of Mahanadi and Godavari basin. High recycled precipitation due to land-atmosphere feedback is well established for the Ganga basin^{32,33,42}. Cauvery basin, on the other

hand, has a large number of incoming links from all other basins, as evident from figure 1(c). Literature shows that the Cauvery basin receives recycled precipitation generated by evapotranspiration from the neighboring regions⁴³. Stronger causal connections exist between the land variables of other basins (figure 1(c), S2). This means that the river basins can have characteristic properties of being 'donor basins' or 'recipient basins' depending on the net transfer of moisture through atmospheric pathways. " (Lines 104 – 114)

"The intra-basin land-to-atmosphere connection happens by SM contributing to the moisture content of the air through heat fluxes (high evapotranspiration during high SM) while causing surface cooling. The LH and SH control the moisture content and moisture flow (through wind, driven by temperature) in the atmosphere. Hence, for upward links (first column in figure 2, S3, and S4) in most of the cases, PCMCi detects LH as the source variable instead of SM. Since SM can be directly impacted by precipitation, we can expect downward links from the atmosphere directly to SM (figure 2, S3 and S4). However, this signal is weak because it is short-lived and dependent on the duration of precipitation spells. Atmospheric variables possess long and sustained controls on soil moisture by modulating latent heat fluxes through winds, clouds, etc. and hence, in case of downward land-atmosphere connections also, links are primarily towards heat fluxes (figure 2, S3 and S4). " (Lines 199 – 209)

Specific comments:

1. Lines 111-113: "... hence, we stick to bivariate analysis." Some information may be missing by reducing the dimensions. Is it possible to analyze the effects of such a particular simplification on the results?

We thank the reviewer for this comment. We stick to bivariate analysis, notwithstanding some information loss due to dimensionality reduction as mentioned by the reviewer, primarily because the TE becomes less robust as the dimensions increase. It is also computationally demanding. The limitation is that we can not distinguish between links that are appearing because of a common driving variable (confounding effect, link between A and C when $A \leftarrow B \rightarrow C$) and links that are serially connected (link between A and C when $A \rightarrow B \rightarrow C$).

Our primary goal in figure 1 is to show that land-land variables across different basins are causally linked, Switching to a multivariate TE model for land variables will not

remove the confounding effect because the atmospheric variables are dominant confounders and are not included in the system. To demonstrate this point, we have now applied PCMCI using its linear estimator (ParCorr) on land variables, and the results (now shown in figure 1(c)) are shown below. We get nearly the same number of links between land variables as TE (shown in supplementary figure S2) because PCMCI cannot condition on true confounders when atmospheric variables are not considered. A complete causal graph can only be captured by including the necessary confounding variables, which we show in figure 2, using non-linear causal delineation PCMCI (CMI-knn). We have now mentioned this in the results and discussions section of the manuscript (Lines 127 – 129 and 141 – 146)

Figure S2: Land-Land connections using Granger Causality (GC, a) and PCMCI (b). Links are shown if they are found statistically significant at 99% confidence.

2. Lines 130-131: "Thus, the most likely reason for these links would be indirect links or common drivers." Any examples?

Since the bivariate estimator of Transfer Entropy (TE) is used, it will be sensitive to the influence of a third variable on the variables under consideration. For example, the El Nino-Southern Oscillation (ENSO) is known to influence the Indian monsoon variability. In such cases, the occurrence of an El Nino reduces the soil moisture in multiple basins simultaneously due to reduced precipitation; a causal delineation technique will show a connection between the soil moisture of two basins which is just

the impact of the confounding factor El Nino. In such a case, the actual connections would be from the El Nino, through its impact on local Precipitation,. It is also possible that the local changes in soil moisture in a river basin may indirectly affect that in another river basin through local land-atmosphere interaction (called an indirect link because the true link from land-atmosphere feedback is getting delineated as a land-land link), which in turn would affect the large-scale flow. The presence of indirect links (or confounders) is confirmed by figure 1(c) generated using the PCMCI as the number of links captured by PCMCI is nearly same as the number of links captured by Granger Causality (GC, now included in response to reviewers' comment) or Transfer Entropy (TE). Since the PCMCI conditions all necessary variables from the input set (methods), it should generate fewer links if confounding variables were present among land variables. Hence, from links shown in figure 1(c), it can be concluded that the variables forming the pathways between land variables need to be included to find true causal linkages between different river basins.

In figure 1, there could be multiple indirect links, for example, as seen in figure 2, there is a link from precipitation in the Ganga basin (P_G) to incoming radiation in the Mahanadi basin (SR_M'), which indicates that moisture from the Ganga basin might contribute to cloud formation in the Mahanadi basin. Precipitation from the Godavari basin (P_Go) is connected to precipitation of the Krishna, Mahanadi, and Narmada-Tapi basins (P_K', P_M', P_NT'). The 4th layer, or the rightmost layer, contains the land variable of the receiving basin. The temperature of the Mahanadi basin (T_M') impacts the latent and sensible heats of the Mahanadi basin (LH_M, SH_M). Hence the pathway from the Ganga basin to the Mahanadi basin (as seen as an indirect link from LH_G to LH_M in figure 1(c)) can be traced as $LH_G \rightarrow P_G \rightarrow SR_M' \rightarrow LH_M$ (in figure 2).

We have now included the above mentioned points in lines 132 – 145 and lines 176 – 184 of new manuscript.

3. Figure 1c: for some variables in some river basins (e.g., SM_K, SM_G, SM_M, and SM_NT), many links are originated from them, but few links are pointed to them. Why? Would you please briefly explain this phenomenon?

We thank the reviewer for the important comment. A typical Link direction in the figure represents the flow of information or causation from a source region to a sink region. Such information transfer is driven by the physical mass or energy transfer between two nodes.

Several links originating from a river basin's soil moisture or latent heat flux indicate that such a basin is one of the major hotspots of land-atmosphere feedback. It drives atmospheric circulation, which in turn, impacts other basin's rainfall and hydrology. This does not of course mean that the source basin's hydrology should also necessarily be driven by land-atmosphere feedback of neighboring basins, as that would depend on the land-atmospheric feedbacks and seasonality of the atmospheric circulation. Hence, it is reasonable that some basins may produce a large number of links but receive few. This means that the river basins have characteristic properties of being 'donor basins' or 'recipient basins' depending on the net transfer of moisture, among other things, through atmospheric pathways. For example, latent heat flux from Ganga basin, LH_G, has outgoing links to almost all river basins, however, it has incoming links only from land variables of Mahanadi and Godavari basin (figure 1(c)). This explanation is added to the new manuscript in the results and discussion section (lines 104 – 114 and 166 – 173)

4. Figure 2: some variables in the donor river basins (e.g., SM_Go, SM_C, and SM_NT) are missing (in the first column), while none of the variables is missing in the receiving river basins (in the fourth column). Any reasons for this? Moreover, I would suggest the authors re-order all the variables according to the basin names, like what they do in Figure S2.

We thank the reviewer for this comment. To improve the readability, and in response to a minor comment by another reviewer, we have now revised figure 2 by using a stricter threshold (more than 50% of links crossing statistical significance at 5% level). We tried to re-order the variables; however, due to software limitations, absolute reordering was not possible. We believe the reviewer (and readers) will find the simplified figure easy to understand.

Missing soil moisture (SM) variables in the first column are expected since the upward feedback from soil moisture to the atmosphere is not direct, and happens through heat

fluxes. Increased SM results in high evapotranspiration, ET (LH), lowering the SH. The LH and SH control the atmospheric moisture content and moisture advection (through their impact on the temperature and thus the wind). Hence, for upward links (first column in figure 2, S3, and S4), in most of the cases, the PCMCI detects LH as the source variable instead of SM. In summary, the signature of the SM presents through the LH and SH and therefore, the SM is missing from the 1st column.

This discussion has been added to the manuscript (lines 198 – 207)

Figure 2: Connections using PCMCI from Land to atmosphere within basin (first column), between atmospheric variables across all basins (second column), and from the atmosphere to land within the basin (third column). A link is shown only if it is found statistically significant at a 5% level more than 50% of the time (20 years out of 40 years (1981-2020)). Names are variable symbols followed by the basin they belong

to, for example, LH_G means latent heat flux from the Ganga basin. The first and second columns of variables are land variables (soil moisture SM, latent heat flux LH, and sensible heat flux SH) and atmosphere variables (precipitation P, temperature T, relative humidity R, wind speed WS, and incoming short wave radiation SR) respectively, links between which represent land-to-atmosphere connections within each basin. Links between the next two columns represent atmosphere-to-atmosphere connections, for example, there is a link from the temperature in the Ganga basin (T_Go) to that of the Mahanadi basin (T_M'). Links in the last column represent downward connections from atmospheric variables to land variables within the basin.

5. Figure 4: why do the authors choose these regions? Are there any special features in these regions? I cannot find related sentences regarding this issue.

We thank the reviewer for this comment and apologise for the confusion. We have added explanations for the selected regions in the manuscript.

The rationale behind this figure is to see if soil moisture from irrigated areas is causally linked to steep reductions in precipitation witnessed in the IRR experiment. We chose these regions from areas containing significant reduction in precipitation (from figure 3, and S5) and the regions where irrigation was applied (figure 1(b)). Figure 4(a) shows three chosen regions (southern peninsula – region A, western India – region B, and a part of Ganga basin – region C) out of areas where irrigation was applied, and figure 4(b) shows a few selected regions where maximum drying was witnessed (Region 1 – central-eastern India, Region 2 – Central India, Region 3 – western India, and Region 4 – western Himalayas). These variables, i.e., LH from regions A, B, and C and P from regions 1, 2, 3, and 4, were taken as differences between CTL and IRR runs. We tried to find the causal links from the latent heat flux (LH, IRR-CTL) of Regions A, B, and C to precipitation (P, IRR-CTL) from regions 1, 2, 3, and 4, using transfer entropy (TE) as we want to capture both direct and indirect connections in model simulations. In principle, irrigation in a region could be causally linked to multiple soil moisture drawdowns as land-atmosphere feedback is complex, and redistributes moisture supply across the Indian subcontinent. We picked the regions where the impact of

river interlinking is maximum (figure 3). This is now explained in the results and discussion section of the manuscript (lines 327 – 337).

6. some typos:

Line 183: "aria" -> "area"

Line 371: "as" -> "has"

Line 383: "iterating" -> "iterates"

Line 411: "Indian monsoon season (ISM; ..."?

We thank the reviewer for pointing out these typos; we have corrected them and thoroughly checked our manuscript for other typos and grammatical errors.

Reviewer 3

1 General Comments

The authors hypothesize in this study that water transfer between river basins may impact the donor or adjacent basins through land-atmosphere feedback. The authors claim that this hypothesis can be tested using causal delineation techniques, a coupled regional climate model, and multiple reanalysis datasets. The study is carried out in India. Moreover, authors also claim that their finding "disputes the conventional assumption of hydrological independence between river basins". The authors indicate that this study is a first attempt (I can't prove not disprove the statement at the moment) to test the hypothesis that large irrigation projects, under the ceteris-paribus-condition, may induce changes in neighboring river basins. To test this hypothesis, the authors use the WRF model under control experiments and analyse the data with two statistical approaches to test statistical causality (i.e., Transfer Entropy and Peter and Clark's Momentary Conditional Independence). Based on the analysis of results, the authors concluded that there is enough statistical evidence to reject the hypothesis that (1) "the hydrology of the river basins is independent" and that (2) "the feedback from interlinking river basins(through water project) will not affect the rainfall patterns in neighboring basins".

The topic of this manuscript may be of interest for the readership of the Nat. Comm. Journal. At the moment, however there are many points that have to be clarified before publication.

2 Specific Comments

This paper has the following technical shortcomings:

- I do not agree with the statement that (L48ff) "the interlinking of the rivers assumes the river basins to be hydrologically independent". It is in my opinion an overstatement. Hydrologically speaking, if two neighboring river basins do not have a common large scale aquifer (e.g. Karstic formation), we can assume that the generated superficial runoff is mostly dependent of the own contributing drainage areas. If there is intercatchment groundwater flows, the river basins are called non-conservative (NC). "There are many well-known physical reasons for the existence of NC catchments. Karstification of limestone is perhaps the most widespread reason worldwide" (Le Moine et al. WRR doi: 10.1029/2006WR005608, 2007). It is widely accepted in the hydrological community that we can't close the water balance only by measurements. Errors in measurements is one cause, the another one is leaking catchments. In south Germany and many basins in France this hypothesis not true (see also Zink et al. HESS 2017). Consequently, a generalization of basins independence is not a valid assumption.

In the present study, the authors have not checked or provide evidence that the basins used are conservative. This should be done.

We thank the reviewer for providing this valuable information and agree that the hydrological independence of the river basins cannot be assumed without considering groundwater flows. However, not all river basins would be linked by inter-catchment groundwater flows, for example, far-off basins like Ganga and Cauvery basins in India. We have removed the statement about hydrological independence from the manuscript and modified all sentences appropriately. We also clarify in the manuscript that our assessment is about the influences of inter-basin water transfer on the surface

water cycle and land-atmosphere interactions, highlighting the influence on groundwater flows as a potential future work:

"This result is at odds with the conventional assumption of absence of atmospheric links between river basins while planning hydrological projects." (Lines 91)

"The assumption behind such planning is that the river basins do not have atmospheric connections, and hence, feedback from linking a basin to other river basins will not affect the rainfall patterns in the source basin." (Lines 370 – 372)

"In addition, the river basins can also exchange water via inter-catchment groundwater flows which is outside the scope of current study." (Lines 403)

• L184ff: 600 mm/day or 1400 mm/day is imposible. Authors probably mean per year. My doubt is confirmed in L424ff! Please correct.

We apologize for this error and thank the reviewer for pointing this out. We have corrected the units in the revised manuscript. (Line 229)

• Please indicate why streamflow, one the key variables in water management, has not been included in this study? In my opinion, it should be included. Authors should also indicate the sign of the feedbacks (e.g. with color).

We thank the reviewer for this comment. If we understand correctly, the reviewer's query is about non-inclusion of stream flow in the causal analysis highlighting land-atmosphere pathways.

The reviewer is correct to point out that stream flow is an important variable for water management. However, the land-atmosphere feedback is primarily driven by basin soil moisture (koster et al 2004, DOI: 10.1126/science.1100217) as basin-wide evaporation is huge compared to evaporation from streamflow alone. The excess or deficit surface runoff can change the soil moisture by irrigation application and hence,

impact land-atmosphere feedback. Since we have already included soil moisture and latent heat flux as variables, we believe that adding streamflow will add minimal new information while increasing the dimensionality multifold as 6 new variables will be introduced (one for each basin, each at 10 lags, hence, total 60 new dimensions).

The sign of the feedback (positive or negative) cannot be shown for figure 2 as the metric used, the conditional mutual information (CMI-knn) in PCMCI does not delineate the sign. It only measures the strength and direction of non-linear association. The PCMCI is capable of capturing the nature of linear association (positive/negative) and we have now shown land-land connections using it in Figure 1(c). Sign associated with each link from PCMCI is shown in supplementary table (ST3) along with the p-value of each link. We observe both positive and negative land atmosphere feedbacks across different river basins. For example, land variables from Ganga have positive links towards land variables of the Mahanadi basin but negative links towards land variables of the Cauvery basin. The existence of both positive and negative connections between different river basins shows that soil moisture from one basin can reduce or amplify soil moisture in another basin. The results and discussion section has been updated to include above mentioned changes (Lines 114 – 119).

ST 3: Details of strengths and p-values of links in network generated of land variables using linear estimator of PCMCI. All values are statistically significant at more than 99% confidence.

Source	Target	Value	pValue	Source	Target	Value	pValue
LH_G	SH_G	0.03845	0	LH_K	LH_Go	0.075393	0
LH_G	SM_G	0.053575	0	LH_K	SH_Go	0.040886	0
LH_G	LH_Go	0.008202	7.04E-36	LH_K	SM_Go	0.0269	0
LH_G	SH_Go	-0.00162	8.56E-14	LH_K	LH_C	0.028471	0
LH_G	SM_Go	-0.00403	1.02E-07	LH_K	SH_C	0.024056	2.4E-135
LH_G	LH_C	-0.00351	3.28E-06	LH_K	SM_C	0.010652	2.97E-29
LH_G	SH_C	-0.00519	1.02E-09	LH_K	SH_K	0.02549	0
LH_G	SM_C	-0.00235	0.001787	LH_K	SM_K	0.032268	0
LH_G	SM_K	0.001853	0.007463	LH_K	LH_M	0.027164	5.3E-227
LH_G	LH_M	0.039875	0	LH_K	SH_M	0.013419	3.41E-48
LH_G	SH_M	0.015474	1.9E-192	LH_K	SM_M	0.007387	4.05E-73
LH_G	SM_M	0.021769	2.1E-149	LH_K	LH_NT	0.010432	1.04E-32
LH_G	SH_NT	-0.00479	1.99E-10	LH_K	SH_NT	0.001028	4.99E-13
LH_G	SM_NT	-0.00875	8.91E-06	LH_K	SM_NT	0.001267	5.84E-06
SH_G	LH_G	0.041838	0	SH_K	SH_G	0.00909	0.000795

SH_G	SM_G	0.052415	0	SH_K	SM_G	0.000134	0.001101
SH_G	LH_Go	0.003006	6.73E-05	SH_K	LH_Go	0.044144	0
SH_G	SH_Go	0.012713	2E-101	SH_K	SH_Go	0.073801	0
SH_G	SM_Go	0.010733	1.14E-58	SH_K	SM_Go	0.052429	0
SH_G	SH_C	-0.00306	5.05E-05	SH_K	LH_C	0.016903	3.3E-113
SH_G	SH_K	0.00253	0.000795	SH_K	SH_C	0.035028	0
SH_G	SM_K	0.002607	0.000547	SH_K	SM_C	0.013849	3.3E-76
SH_G	LH_M	0.017947	1.2E-127	SH_K	LH_K	0.025688	0
SH_G	SH_M	0.04151	0	SH_K	SM_K	0.061566	0
SH_G	SM_M	0.031608	0	SH_K	LH_M	0.021765	4.2E-120
SH_G	LH_NT	-0.00347	4.11E-06	SH_K	SH_M	0.030191	9.2E-251
SM_G	LH_G	0.042841	0	SH_K	SM_M	0.021133	1.3E-177
SM_G	SH_G	0.06515	0	SH_K	LH_NT	0.009426	6.82E-12
SM_G	LH_Go	0.008781	6.75E-08	SH_K	SH_NT	0.012538	1.22E-62
SM_G	SH_Go	0.006157	1.42E-13	SH_K	SM_NT	0.002238	2.44E-09
SM_G	SM_Go	0.012202	2.1E-59	SM_K	LH_G	-0.00202	0.007463
SM_G	LH_C	-0.00247	0.00105	SM_K	SH_G	0.002607	0.000547
SM_G	SM_C	-0.00195	0.00989	SM_K	SM_G	0.002343	0.001888
SM_G	SH_K	0.005498	0.001101	SM_K	LH_Go	0.028901	9.4E-207
SM_G	SM_K	0.002343	0.001888	SM_K	SH_Go	0.039372	0
SM_G	LH_M	0.020266	3.9E-163	SM_K	SM_Go	0.065198	0
SM_G	SH_M	0.024073	1.1E-231	SM_K	LH_C	0.021687	2.4E-139
SM_G	SM_M	0.049947	0	SM_K	SH_C	0.029828	1.8E-191
SM_G	LH_NT	-0.00152	2.51E-07	SM_K	SM_C	0.024644	8.3E-154
SM_G	SH_NT	-0.00098	3.71E-15	SM_K	LH_K	0.031786	0
LH_Go	LH_G	0.009405	7.04E-36	SM_K	SH_K	0.076832	0
LH_Go	SH_G	-0.00021	6.73E-05	SM_K	LH_M	0.002869	2.89E-18
LH_Go	SM_G	0.000261	6.75E-08	SM_K	SH_M	0.010551	4.74E-67
LH_Go	SH_Go	0.03493	0	SM_K	SM_M	0.023426	1.2E-131
LH_Go	SM_Go	0.028114	0	SM_K	SH_NT	-0.00021	4.85E-09
LH_Go	LH_C	0.005083	3.91E-34	SM_K	SM_NT	0.002815	7.61E-12
LH_Go	SH_C	0.008318	2.11E-28	LH_M	LH_G	0.043292	0
LH_Go	LH_K	0.065417	0	LH_M	SH_G	0.015169	1.2E-127
LH_Go	SH_K	0.029566	0	LH_M	SM_G	0.02212	3.9E-163
LH_Go	SM_K	0.020232	9.4E-207	LH_M	LH_Go	0.057274	0
LH_Go	LH_M	0.044525	0	LH_M	SH_Go	0.039454	0
LH_Go	SH_M	0.021148	2.8E-242	LH_M	SM_Go	0.026298	5.4E-171
LH_Go	SM_M	0.009295	3.1E-157	LH_M	LH_C	0.004493	0.003138
LH_Go	LH_NT	0.043336	0	LH_M	SH_C	0.002804	0.0002
LH_Go	SH_NT	0.016929	1.42E-73	LH_M	SM_C	-0.00299	7.25E-05
LH_Go	SM_NT	0.015563	4.41E-57	LH_M	LH_K	0.022304	5.3E-227
SH_Go	LH_G	0.002689	8.56E-14	LH_M	SH_K	0.025918	4.2E-120
SH_Go	SH_G	0.01612	2E-101	LH_M	SM_K	0.009373	2.89E-18
SH_Go	SM_G	0.00155	1.42E-13	LH_M	SH_M	0.039579	0

SH_Go	LH_Go	0.037102	0	LH_M	SM_M	0.044798	0
SH_Go	SM_Go	0.053536	0	LH_M	SH_NT	-0.00054	2.16E-05
SH_Go	LH_C	0.003973	9.33E-20	LH_M	SM_NT	-0.00306	0.000161
SH_Go	SH_C	0.012983	6.59E-51	SH_M	LH_G	0.024854	1.9E-192
SH_Go	SM_C	0.00218	0.003844	SH_M	SH_G	0.046888	0
SH_Go	LH_K	0.031531	0	SH_M	SM_G	0.017122	1.1E-231
SH_Go	SH_K	0.065485	0	SH_M	LH_Go	0.027228	2.8E-242
SH_Go	SM_K	0.047489	0	SH_M	SH_Go	0.057149	0
SH_Go	LH_M	0.02809	0	SH_M	SM_Go	0.033856	0
SH_Go	SH_M	0.052242	0	SH_M	SH_C	0.002455	0.001133
SH_Go	SM_M	0.030827	0	SH_M	LH_K	0.002383	3.41E-48
SH_Go	LH_NT	0.016412	1.1E-106	SH_M	SH_K	0.02778	9.2E-251
SH_Go	SH_NT	0.033789	0	SH_M	SM_K	0.010847	4.74E-67
SH_Go	SM_NT	0.016505	7E-108	SH_M	LH_M	0.039967	0
SM_Go	LH_G	0.004013	1.02E-07	SH_M	SM_M	0.055368	0
SM_Go	SH_G	0.013175	1.14E-58	SH_M	SM_NT	-0.00637	0.009127
SM_Go	SM_G	0.012409	2.1E-59	SM_M	LH_G	0.019404	2.1E-149
SM_Go	LH_Go	0.042057	0	SM_M	SH_G	0.02976	0
SM_Go	SH_Go	0.071504	0	SM_M	SM_G	0.043234	0
SM_Go	LH_C	0.008234	7.49E-28	SM_M	LH_Go	0.0352	3.1E-157
SM_Go	SH_C	0.018989	1.93E-58	SM_M	SH_Go	0.047974	0
SM_Go	SM_C	0.009964	5.92E-13	SM_M	SM_Go	0.059012	0
SM_Go	LH_K	0.026305	0	SM_M	LH_C	0.00293	0.000102
SM_Go	SH_K	0.050647	0	SM_M	SH_C	0.011167	5.16E-09
SM_Go	SM_K	0.07094	0	SM_M	LH_K	0.002092	4.05E-73
SM_Go	LH_M	0.017135	5.4E-171	SM_M	SH_K	0.033528	1.3E-177
SM_Go	SH_M	0.030503	0	SM_M	SM_K	0.027154	1.2E-131
SM_Go	SM_M	0.045775	0	SM_M	LH_M	0.041453	0
SM_Go	LH_NT	0.009681	8.94E-22	SM_M	SH_M	0.066579	0
SM_Go	SH_NT	0.013999	1E-104	SM_M	LH_NT	-0.00179	1.6E-10
SM_Go	SM_NT	0.033933	7.9E-233	SM_M	SH_NT	-0.00353	2.24E-13
LH_C	LH_G	-0.00351	3.28E-06	SM_M	SM_NT	-0.0097	2.46E-05
LH_C	SM_G	-0.00021	0.00105	LH_NT	SH_G	-0.00937	4.11E-06
LH_C	LH_Go	0.012732	3.91E-34	LH_NT	SM_G	-0.00784	2.51E-07
LH_C	SH_Go	0.013788	9.33E-20	LH_NT	LH_Go	0.037204	0
LH_C	SM_Go	0.008234	7.49E-28	LH_NT	SH_Go	0.016412	1.1E-106
LH_C	SH_C	0.043582	0	LH_NT	SM_Go	0.010145	8.94E-22
LH_C	SM_C	0.04244	0	LH_NT	SM_C	-0.00211	0.005233
LH_C	LH_K	0.034419	0	LH_NT	LH_K	0.009738	1.04E-32
LH_C	SH_K	0.019767	3.3E-113	LH_NT	SH_K	0.004649	6.82E-12
LH_C	SM_K	0.018802	2.4E-139	LH_NT	SM_M	-0.01023	1.6E-10
LH_C	LH_M	0.002228	0.003138	LH_NT	SH_NT	0.024994	0
LH_C	SM_M	0.00293	0.000102	LH_NT	SM_NT	0.035754	0
LH_C	SH_NT	0.002058	0.006357	SH_NT	LH_G	0.000545	1.99E-10

SH_C	LH_G	-0.00734	1.02E-09	SH_NT	SM_G	-0.01294	3.71E-15
SH_C	SH_G	-0.00306	5.05E-05	SH_NT	LH_Go	0.018121	1.42E-73
SH_C	LH_Go	0.019975	2.11E-28	SH_NT	SH_Go	0.04611	0
SH_C	SH_Go	0.022388	6.59E-51	SH_NT	SM_Go	0.022634	1E-104
SH_C	SM_Go	0.015912	1.93E-58	SH_NT	LH_C	0.004744	0.006357
SH_C	LH_C	0.044699	0	SH_NT	LH_K	0.005445	4.99E-13
SH_C	SM_C	0.055946	0	SH_NT	SH_K	0.010164	1.22E-62
SH_C	LH_K	0.031875	2.4E-135	SH_NT	SM_K	0.004411	4.85E-09
SH_C	SH_K	0.03927	0	SH_NT	LH_M	-0.0032	2.16E-05
SH_C	SM_K	0.024862	1.8E-191	SH_NT	SM_M	-0.00354	2.24E-13
SH_C	LH_M	0.004838	0.0002	SH_NT	LH_NT	0.028226	0
SH_C	SH_M	0.005101	0.001133	SH_NT	SM_NT	0.060615	0
SH_C	SM_M	0.006755	5.16E-09	SM_NT	LH_G	-0.00335	8.91E-06
SM_C	LH_G	-0.00235	0.001787	SM_NT	LH_Go	0.011252	4.41E-57
SM_C	SM_G	-0.00195	0.00989	SM_NT	SH_Go	0.012878	7E-108
SM_C	SH_Go	0.00218	0.003844	SM_NT	SM_Go	0.023938	7.9E-233
SM_C	SM_Go	0.003293	5.92E-13	SM_NT	LH_K	-0.00271	5.84E-06
SM_C	LH_C	0.041623	0	SM_NT	SH_K	-0.00626	2.44E-09
SM_C	SH_C	0.067336	0	SM_NT	SM_K	-0.00043	7.61E-12
SM_C	LH_K	0.005184	2.97E-29	SM_NT	LH_M	-0.00285	0.000161
SM_C	SH_K	0.016173	3.3E-76	SM_NT	SH_M	-0.00197	0.009127
SM_C	SM_K	0.031557	8.3E-154	SM_NT	SM_M	0.001582	2.46E-05
SM_C	LH_M	-0.00793	7.25E-05	SM_NT	LH_NT	0.030938	0
SM_C	LH_NT	-0.00722	0.005233	SM_NT	SH_NT	0.051032	0

• I wonder why the authors excluded the well known approach proposed by Granger. They indicate that the TE and the PCMCi capture non-linearities, but there is a price for that: the increase of dimensionality and computational time. If the Granger Causality test is rejected, the authors could investigate non-linear causality pathways. I suggest the present a map of the p-values of the H_0 : X_t is not Granger causing Y_t . Instead of trying to find all potential connection, I would attempt to find the most obvious ones based on literature. Or to confirm the strong ones found by these two methods.

We thank the reviewer for this valuable comment. In response, we have performed Granger causality analysis on land variables before performing the Transfer Entropy analysis.

We agree with the reviewer that performing non-linear causal delineation comes with a cost and starting with a linear test would be a better idea. Hence, we have now

included Granger Causality (GC) and PCMCI with its linear estimator (ParCorr) in our methods. We first compute causal connections between land variables using pairwise GC and then apply bivariate TE to highly statistically significant connections found GC. Since both GC and TE used in this analysis are bivariate, we also used the linear estimator of PCMCI (ParCorr) which finds linear causal associations while controlling for high dimensionality.

We have updated methods section and added a description of Granger Causality (GC) in Lines 431 – 448: "Granger causality (GC) is a causal delineation technique based on two conditions: first, that cause must precede the action, and second, cause must hold predictive information for the effect. A time series \mathbf{X} can be argued to 'granger cause' a time series \mathbf{Y} if past of \mathbf{X} can add unique predictive information for \mathbf{Y} when the predictive information from the past of \mathbf{Y} has already been accounted for. First, we regress the time series \mathbf{Y} with its past up to some lag τ followed by addition of past values of \mathbf{X} as regressors (also called vector autoregression (VAR) models).

$$Y_t = c + \sum_{l=1}^{\tau} a_l Y_{t-l} + \varepsilon_t$$

$$Y_t = c + \sum_{l=1}^{\tau} a_l Y_{t-l} + \sum_{l=1}^{\tau} b_l X_{t-l} + \eta_t$$

Here τ is the maximum lag, a and b are the regression coefficients, and ε and η are noise terms. If the second regression model is a better fit than the first one, then \mathbf{X} can be argued to 'granger cause' \mathbf{Y} . In this study, we perform pairwise GC analysis using VAR models of maximum lag 10 to test causal connections between land variables of different river basins. The best fit models were classified based on Akaike Information Criterion (AIC). We present a network using a binary adjacency matrix which contains value 1 at any i^{th} row and j^{th} column if i^{th} variable granger found to granger cause j^{th} variable at 99% statistical significance. The major assumptions of GC are that the data is stationary and can adequately be described using a linear model."

The land to land connections found using pairwise GC at 99% confidence level are shown in supplementary figure S2(a)(also shown below). Our GC analysis shows that all variables are connected to one other at 99% confidence level. Next, we compute

TE between these variable pairs and results are shown in supplementary figure S2(b). The TE is argued to be a non-linear extension of GC (Barnett et al 2009, <https://doi.org/10.1103/PhysRevLett.103.238701>). We observe slightly lesser number of links using TE (in comparison to GC) as while conditioning it is able to remove links appearing because of strong non-linear autocorrelations. Since both of these methods are bivariate, they don't account for confounding effects of other variables within inputs or indirect links. Performing multivariate GC and TE is infeasible as it will increase the dimensionality multifold and will lead to significant loss of statistical power which increases the chances of getting spurious links. To address that, we used PCMCI with its linear estimator (ParCorr) to find causal connections between land variables as it is able to reduce dimensionality of the conditioning set while controlling for spurious causalities. The results are now shown as figure 1(c). PCMCI (ParCorr), even after considering multiple variables in conditioning set still shows almost equal number of links as TE which proves that the confounding variables of land variables are present outside the system. This sets the rationale for including atmospheric variables in the causal analysis. The results and discussion section is now updated to include PCMCI as primary figure 1(c) with GC and TE added as supplementary figure 2 (a,b). The discussion around land-land connections has also been updated (Lines: 99 – 129).

For generating figure 2, GC failed because of high covariance between the same variables across different basins. PCMCI can handle this caveat. Hence, we chose to perform analysis using PCMCI for figure 2.

Figure S2: Land-Land connections using Granger Causality (GC, a) and Transfer Entropy (TE, b). Links are shown if they are found statistically significant at 99% confidence.

Figure 1: (a) River Basins in India considered in the study. (b) Irrigated grid cells under river interlinking schemes showing change in percentage irrigated area from CTL to IRR run (see methods) to increase irrigated area fraction to 80%. (c) Network between land variable across river basins generated using PCMCI (ParCorr). Sectors are labelled as variable symbols (soil moisture-SM, latent heat flux-LH, sensible heat flux-SH) followed by the basin they belong to (Ganga (G, 808334 km²), Godavari (Go, 302063 km²), Mahanadi (M, 139659 km²), Krishna (K, 254743 km²), Narmada-Tapi (NT, 98,796 km², 65,145 km² respectively – two river basins taken together), and Cauvery (C, 85624 km²)). Links are only shown if found statistically significant at 99% confidence and are colored same as the node they originate from.

For example, link from LH_G to LH_M shows that there is a connection between latent heat fluxes from Ganga and Mahanadi basin. Ratio of incoming to outgoing links in Cauvery basin is very high compared to Ganga and Narmada-Tapi basin.

• I also don't understand why two methods are presented (TE and the PCMCI). What is the role of TE and PCMCI? Do both methods find the same causation pathways? Not clear for me. The authors should present the same kind of graph depicting the links. In Figure 4 I don't understand what the width of the link means? What is the statistical significance in Fig 4, and elsewhere?

We thank the reviewer for this valuable comment. In response, we have now changed figure 1 to shown networks using PCMCI and added GC and TE to supplementary figure S2.

Figure 1 aims to show that pairwise causalities exist between land variables across different basins. Since these direct connections are not possible as the physical transfer of water does not take place from land to land, the presence of these links indicates that either there is an indirect link between two land variables (physical transfer happening indirectly) or there is a common driver, such as ENSO, which affects both of these variables. To show consistency across methods, we have now generated land-land network using PCMCI (ParCorr), shown in figure 1, and with Granger Causality (GC) and Transfer Entropy (TE), presented in supplementary figure S2. PCMCI also gives a very high number of links because, for eliminating indirect/confounding links, the confounders should be present in the system, which is not the case for land-land variables as major confounding variables are outside the system (atmospheric variables).

Hence, to examine the possibility of indirect links and eliminate the possibility of common drivers, we use atmospheric variables along with land variables in PCMCI, as it can remove the impacts of common drivers in large dimensional datasets while controlling for false causal discovery. This step cannot be done with TE or GC as the robustness of multivariate analysis is low in high-dimensional datasets.

In figure 4, the link thickness and the link labels at the beginning of the link show the number of times the link was found statistically significant out of 22 years of simulations. We have counted only those links which were found statistically significant at 95% confidence level. We have now updated the figure caption and the manuscript results section to include the information about significance level. Mentioning individual p-value is difficult as each link has 22 different p-values and we count the number of times we find it statistically significant at 95% confidence level.

We have now ensured mentioning of statistical significance levels everywhere in the manuscript. (Lines 101, 122, 156, 237, 262, 263, 342, 351, and all figure captions)

Figure S2: Land-Land connections using Granger Causality (GC, a) and PCMCI (b). Links are only shown if found statistically significant at 99% confidence level.

• The authors use very sophisticated methods to investigate causal relationships but I can't read anywhere the p-values of the most important causal pathways. If there exist, p-values ≤ 0.05 for the links shown in Fig 2, why not to depict them?

We thank the reviewer for this comment. To improve readability, we have now generated figure 2 using a stricter threshold. We have shown only those links which

were found statistically significant at 5% significance level ($p < 0.05$) and are present in at least 50% of the years. This is now mentioned in each figure caption and also in lines 101, 122, 156, 237, 262, 263, 342, 351 in the results and discussion section. Since figure 2 is generated from an ensemble of networks, individual p-values are different for each link across different years i.e. each link has 40 p-values, hence, showing each p-value will be difficult. We count the number of times the link was statistically significant out of 40 years ($p < 0.05$) and consider the link in the average network only if the count is more than 50% of samples (20 out of 40 years).

• There are several publications in which it has been tested that irrigation "can affect" local climate (e.g., <https://gmd.copernicus.org/articles/13/3179/2020/gmd13-3179-2020.pdf>). In this study, for example, the authors concluded that "Recent literature shows that irrigation mostly affects the near-surface variables, creating the so-called irrigation cooling effect.". It is advisable that the authors perform several sensitivity experiments with WFT and its several parameterization mechanisms to be able to conclude that these causation pathways are not simple artifacts of a small sample.

We thank the reviewer for sharing the literature, and agree that the sensitivity to irrigation parameterisations and the amount of water applied are important considerations for regional simulations.

The study shared by the reviewer (Valmassoi et al. 2020) highlights the importance of using realistic irrigation practices to improve regional simulations and proposes three different irrigation parameterisations for WRF. Since, flood irrigation is mostly practiced in India (Yadav et al. 2017, Singh et al. 2021, Water and Agriculture in India by GFFA 2017), these irrigation parameterisations performed poorly in simulating Indian monsoon as none of these parameterisations considered flood irrigation (demonstrated in prior work, Devanand et al 2019; <https://doi.org/10.1029/2019GL083875>). We had examined changes in land-atmosphere feedback with varying irrigation parameterisation in our model set-up in prior work (documented in Devanand et al. 2019), and found that it is necessary to represent irrigation practices specific to India to capture the intraseasonal characteristics of

monsoon precipitation. Devanand et al. 2019 generated India specific irrigation parameterisation for WRF which included flood irrigation and has been demonstrated to perform well in simulating Indian monsoon (Devanand et al. 2019 — doi.org/10.1029/2019GL083875, Jha et al. 2022 — doi.org/10.1038/s41467-022-31962-5). In this manuscript, we utilise the same model-setup with India-specific irrigation practices to apply extra water amounts according to the water transfer targets proposed in the Detailed Project Reports (DPRs; Reference 24) of the interlinking projects by the government of India. Hence, the model used in this study has already been tested for several irrigation parameterisations and contains the best suited irrigation parameterisation for Indian scenario. We believe that this model set-up is well suited to study the response of land-atmosphere interactions to inter-basin water transfer as explained below. In addition, we would also like point out that the cooling effect of irrigation is also captured by our model simulations shown in supplementary figure S6 (a) where we observe a reduction in maximum daily temperatures at locations where extra irrigation has been applied. We have now included this information in the Data and Methods section of the manuscript (lines 532 – 535 and 568 – 571).

Further, to make sure that the links are not just artifacts of the sample and are actual characteristics of the system, and to solidify our contention about the existence of causal pathways, we apply the procedure to identify causal pathways in the regional climate model control runs (WRF-CTL) and show the results in Supplementary Figure S4. The results show that causal pathways similar to that in the reanalysis datasets exist in model runs. We have included this information in the results and discussion section of the manuscript (Line 188 – 199).

Supplementary Figure S4. Causal connections of land-atmosphere interactions across different basins in WRF-CTL runs using PCMCi (similar to Figure 2). A link is shown only if it is found statistically significant with $p < 0.05$ more than 50% of the time (11 years out of 22 years).

• The authors do not report confidence levels in any of the percentage increases (e.g., L209. This is a must in my opinion.

We thank the reviewer for this comment. This is an important point, and we acknowledge that we missed it in our submission. We have now replaced the bar plots with violin plots for each region which gives more information about the distribution of the precipitation change. Mean and median values are also marked for each region. To test for the statistical significance of mean precipitation change, we performed a

paired-sample t-test on September precipitation from CTL and IRR runs (660 data points). p-values are now written under each violin plot (figure 3).

We have now modified the discussion around figure 3 in the manuscript to mention the statistical significance and include the distributions from violin plots to generate inferences. The results section now reads: "Figure 3 (a) shows the spatial distribution of mean percentage change in September rainfall for various regions in India for all years. Statistical significance of mean precipitation change was tested using a t-test on September precipitation from CTL and IRR runs (660 data points) and statistically significant regions have been hatched in the figure. The regions which experience statistically significant ($p < 0.05$) declines in September monsoon rainfall are central India (R1), east peninsular India (R2), coastal Gujrat (R3), east central India (R4), dry western region of Rajasthan (R5), western Himalayan foothills in Uttarakhand (R6). Figure 3(b) shows the probability distribution of percentage change in these regions as violin plots with the median and mean value shown by the red and black colored horizontal lines, respectively, and p-values of the t-test on mean precipitation change written below each violin plot. It is worth noting that the distribution of precipitation changes in these regions contains huge tails on positive values, particularly R3, R4, R5, R7, and R8. They experience sporadic events of precipitation increase in the form of extreme precipitation (100-200% increase) and a systematic precipitation decrease with most of the values in the distribution being negative. Due to this, while a region experiences multiple years of huge precipitation deficits, we underestimate the precipitation reduction due to the heavy tails of surplus years. For example, in R3, R4, and R5, the precipitation reduction with the highest frequency is somewhere between 20-30%, whereas the mean value in these regions reports negligible or a positive change which could be misleading. Hence, while the driest region of India, the western region of Rajasthan (figure 1, R4), shows a mean increase in September precipitation of around 10%, it actually experiences a 10% median decline in precipitation. The highest median reduction is around 12 % in region R4 (the state of Odisha) followed by 10% in R2 (the state of Andhra Pradesh), 10% in R5 (the state of Rajasthan), and 9% in R3 (the state of Gujarat). The rainfall in central India, a part of the core monsoon zone, also shows an 8% decline in the simulated September rainfall due to interlinking. The western Himalayan foothills in Uttarakhand and east-central India also show a moderate decline (6.4%) in September rainfall due to excess irrigation from the

proposed interlinking. It is worth noting that while there is a reduction in September precipitation in generally dry parts of the country, there is also an increase in September precipitation by up to 12% in east India (R8, states of Bihar, Jharkhand, and eastern Uttar Pradesh) and up to 10% in parts of the Deccan plateau (R7, states of Maharashtra and Telangana)." (Lines 259 – 288)

- I do not fully understand the experiment carried out by the authors. For example: 1) how many WRF simulations were carried out by the authors to account for parameter uncertainty and initialisation. 2) If a pathway is found in a given period of time, the authors should demonstrate that this pathway also occur in other locations or periods of time. 3) Authors should indicate how big is the sample size (ensamble of simulations or initializations).

We thank the reviewer for raising this.

We perform a model simulation that includes realistic representation of irrigation practices in India utilising a WRF model configuration that has shown reasonable skill in simulating the Indian monsoon, in addition to several experiments to test forsensitivity to initial conditions. We elaborate on these modelling choices below.

In response to points (1) and (3) raised by the reviewer, we have now included four additional simulations to assess the differences in precipitation with changes in model initialisation, and find that initialization 15-days prior to the monsoon season is sufficient. We performed simulations at varying spin ups of 15, 30, 45, and 60 days for the year 2000 and the results have been added in supplementary figure S10 (also

added below this answer). The results show that the model spin up period is sufficient to generate reliable estimates of precipitation during ISM. To clarify this, we have updated the methods section (Lines 573 – 583)

The model setup that we use has been evaluated for irrigation parameterisation uncertainty in prior work (Devanand et al. 2019). In the present study, we utilise the same model set-up with realistic irrigation practices to apply extra irrigation to satisfy the targets of the planned inter-basin water transfer schemes. We have now clarified these points in the manuscript (Lines 568 – 571).

In response to point (2) raised by the reviewer, we would like to clarify that while presenting networks (figures 1(c), 2, S2, S3, S4, and S9), only those links are considered robust, which are found statistically significant at 95% confidence level (99% for figure 1(c) and S2) above 50% of the time (20 out of 40 years for reanalysis and 11 out of 22 years for model simulations). For results generated from WRF simulations, we check for statistical significance of the difference between two simulations using a t-test and mark the regions which were statistically significant at 90% confidence level. In this way, spatiotemporal consistency is already addressed in our results of causal networks and model simulations. We have also mentioned information regarding statistical significance as well as robustness of links in all figure captions as well as the main text.

S10: Precipitation from WRF CTL runs for the year 2000 with varying spin-ups of 15, 30, 45, and 60 days. A spin-up of 60(45) days means WRF-CTL run was initialized from 1st (15th)April 2000.

- **I suggest to present cross-validation experiments to demonstrate that a causation pathways is not an artifact of a sample but a characteristic of the system. I don't see this at the moment.**

We thank the reviewer for this useful comment and in response, generate causal networks from WRF simulations.

We have now included causal analysis on model simulations (WRF-CTL) in supplementary figure S4. The presence of statistically significant causal pathways across two reanalysis datasets and model simulations provides high confidence that causal links are a characteristic of the system and not just artifacts of sample. The pathways across two datasets and model simulations agree with each other. We have updated the manuscript to include this information. (Lines 188 – 199).

3 Final Remarks

Based on the comments mentioned above and bearing in mind the publishing standards for a research article in a Nature Journal, I recommend to return this paper to the Authors for major revisions.

REVIEWER COMMENTS

Reviewer #2 (Remarks to the Author):

Thank you for addressing my previous comments. Reading this revised manuscript, I have further comments for your referenc.

1. Lines 85-87 mentioned that, through the land-atmosphere feedback, the additional irrigation brought about by river interconnection will lead to changes in the spatial pattern of the Indian summer monsoon. However, the full text does not fully discuss this hypothesis.
2. Lines 154-157: I would suggest the authors create a schematic diagram and try to represent the possible relationships between different variables as clearly as possible.
3. Lines 323-326 summarized the relationship between River interlinking and Indian summer monsoon; however, the causal relationship between River interlinking and Indian summer monsoon needs further clarification.
4. Supporting information: the format of p value in ST 3 is not acceptable.
5. As far as I know, the prerequisites for PCMCI are three basic assumptions, i.e., causal sufficiency assumption, causal Markovian assumption, and loyalty assumption. The sequence is also required to satisfy the stationarity assumption. However, the authors did not make a judgment on the causality between variables. Since I don't know the correlation between different watersheds in India, I have no way of knowing how some physical variables are causal between different watersheds. For example, some physical variables are correlated between Ganga and Cauvery watersheds; however, they are too far. If the result of the algorithm is only judged by the significance, will it cause false positive results? Moreover, the authors did not mention the stationarity of these sequences. If the sequence is non-stationary in future, can we consider the authors' judgment to be unreliable?

Reviewer #3 (Remarks to the Author):

I would like to thank the authors for the detailed explanations and inclusion of the GC plot. It is very enlightening to see the comparison with PCMCI (b). All other questions have been clarified and I am satisfied with the answers.

L.Samaniego.

We thank the reviewer for the insightful comments. We have given a point-by-point response to all the reviewer's comments and made necessary changes to the manuscript in response to suggestions by reviewer. We believe that the comments provided have significantly improved the quality of our manuscript.

Reviewer comments are shown in bold. Author responses are shown in plain text.

#Reviewer 2

Thank you for addressing my previous comments. Reading this revised manuscript, I have further comments for your reference.

1. Lines 85-87 mentioned that, through the land-atmosphere feedback, the additional irrigation brought about by river interconnection will lead to changes in the spatial pattern of the Indian summer monsoon. However, the full text does not fully discuss this hypothesis.

We thank the reviewer for pointing this out. We acknowledge that while our results are centered around the mentioned hypothesis, we did not discuss it verbatim in the manuscript. We have now modified the results and discussion sections to include the hypothesis. Now lines 222-226 discuss the hypothesis in the results, and lines 292-297 and 392-395 lead to its conclusion in the manuscript.

2. Lines 154-157: I would suggest the authors create a schematic diagram and try to represent the possible relationships between different variables as clearly as possible.

We thank the reviewer for this excellent suggestion. As suggested, we have generated a schematic diagram (shown below) that represents connections between different river basins. The schematic is made in such a way that it conveys the problem statement in a simplified manner. The generated schematic, now added as figure 3 in the manuscript, will help communicate the results to larger audiences. It conveys that the perturbations in the land water management leading from the inter-basin water transfer impact the spatial pattern of rainfall on the distant basins. The intra-basin land-to-atmosphere connection happens in the form of SM, contributing to the moisture content of the air through evapotranspiration (high evapotranspiration during high SM) while also causing surface cooling. The supplied moisture by evapotranspiration can lead to recycled precipitation in the same basin or can get transported to faraway regions by the wind, which can then change the precipitation patterns of the region. Evaporative cooling changes the thermal contrast between ocean and land or in between different land regions changing wind patterns and, subsequently, the moisture transport and rainfall. We have also updated the results and discussion section of the manuscript (lines 198-219) to discuss the land-atmosphere pathways of between river basins through the schematic diagram.

Figure 3: Schematic diagram explaining the land-atmosphere feedback and changes in monsoon rainfall in response to river-interlinking.

3. Lines 323-326 summarized the relationship between River interlinking and Indian summer monsoon; however, the causal relationship between River interlinking and Indian summer monsoon needs further clarification.

We thank the reviewer for this comment. Two arguments from model simulations already establish the causal relationship between interlinking and the Indian monsoon and have been discussed in the manuscript. We elaborate on them to clarify the confusion.

First, our model's CTL and IRR runs only differ in the presence of irrigation applied by river interlinking; hence, any change from CTL to IRR must have the river interlinking as its cause.

Second, to check if the reported changes in ISM precipitation are causally linked to the regions where extra irrigation was applied, we generate figure 5 (figure 4 in the previous submission), showing causal links from irrigated regions to regions with changed precipitation demonstrating causal connections from interlinking to the Indian monsoon. It shows that the extra irrigation applied in river interlinking causally drives precipitation change in India during monsoon.

While the exact source-target relationships of land-atmosphere feedback are difficult to delineate from model simulations and are left as a future scope of this work, the above two arguments are sufficient to conclude that river interlinking has causal connections to the Indian summer monsoon.

We have now modified the results and discussion section to clarify these points. (Lines 339-357)

4. Supporting information: the format of p-value in ST 3 is not acceptable.

We thank the reviewer for pointing this out. We have corrected the representation of p-values in ST3.

5. As far as I know, the prerequisites for PCMCI are three basic assumptions, i.e., causal sufficiency assumption, causal Markovian assumption, and loyalty assumption. The sequence is also required to satisfy the stationarity assumption. However, the authors did not make a judgment on the causality between variables. Since I don't know the correlation between different watersheds in India, I have no way of knowing how some physical variables are causal between different watersheds. For example, some physical variables are correlated between Ganga and Cauvery watersheds; however, they are too far. If the result of the algorithm is only judged by the significance, will it cause false positive results? Moreover, the authors did not mention the stationarity of these sequences. If the sequence is non-stationary in the future, can we consider the authors' judgment to be unreliable?

We thank the reviewer for this valuable comment, and in response, we discuss the reliability of PCMCI, perform tests for stationarity, and discuss limitations in the manuscript.

If we understand correctly, the reviewer's concern is that given the assumptions of PCMCI and an assumption of stationarity, how reliable are the inferred causal links between far-away river basins even though we are performing hypothesis tests at a high statistical significance?

Reviewer's concern is correct that even a statistically significant link can be spurious if the assumptions mentioned don't hold true for the data because, in all causal delineation techniques, the fulfillment of causal sufficiency is left to the researcher's judgment. Earth science datasets become even more difficult to handle because systems are complex, with multiple interacting components and no boundaries. In our study, we believe that taking three land variables and five atmospheric variables from each basin nears causal sufficiency for land-atmosphere interactions. Primary processes governing land-atmosphere interactions are evapotranspiration (which supplies the moisture), advection (winds advect moisture to nearby/far-away places), and condensation (which then precipitates that moisture onto distant lands). These processes are well documented in the literature (<https://doi.org/10.1016/j.earscirev.2010.02.004>). Our study includes all the necessary variables reported in the literature that drive these three processes, and causal links between different river basins also capture the advection of moisture to nearby/faraway regions. Some external confounders like ENSO are not included because they bring about low frequency (interannual) variability in land-atmosphere processes, and since we performed the analysis for every year separately, including low-frequency external confounders is not needed. Hence, our dataset satisfies the primary assumptions of PCMCI to the best of our knowledge. Still, there is no denying that the data might violate other assumptions, for example, the assumption of independence of noise terms, which can also lead to spurious links. Hence, we have now

added a paragraph explaining limitations related to assumptions in the methods section of the manuscript (Lines 557-578).

While we understand the reviewer's concern regarding the reliability of a causal link between far-away basins, in our statistical causal analysis, each link goes through a two-stage robustness check. First, it is tested for statistical significance within PCMCi at a 95% confidence level. Next, once statistically significant links are found for each year, a link is considered robust and reliable if found more than 50% of the time. This two-stage robustness check strengthens the reliability of our judgment on a link between two river basins. In addition, we validate our results from another reanalysis dataset and WRF model simulations (again using the above-mentioned two-stage robustness check) which further solidifies our argument. Moreover, the literature contains conclusive evidence for the presence of non-local impacts of land-atmosphere feedback. For example, Meehl (1994) ([https://doi.org/10.1175/1520-0442\(1994\)007<1033:IOTLSI>2.0.CO;2](https://doi.org/10.1175/1520-0442(1994)007<1033:IOTLSI>2.0.CO;2)) demonstrated positive and negative feedback of soil moisture on local and non-local monsoon precipitation. Works of Pathak et al. (2014) (<https://doi.org/10.1175/JHM-D-13-0172.1>) and Paul et al. (2018) (<https://doi.org/10.1029/2018GL078198>) have demonstrated that evapotranspiration in one region can lead to precipitation in faraway regions of the Indian subcontinent. Results from BOLLASINA et al. (2011) (<https://doi.org/10.1126/science.1204994>) show that aerosol forcing and subsequent land cooling is responsible for the reduction in precipitation over north central India. Excess irrigation also results in similar cooling and a possible decline in monsoon rainfall. Paul et al. (2016) (<https://doi.org/10.1038/srep32177>) show that LULC changes can weaken the Indian summer monsoon through declining recycled precipitation. Irrigation also has significant contributions to recycled precipitation. The resultant of irrigation-induced cooling and increased recycling may have complex impacts on monsoon rainfall, which we study in the present work through regional modeling. Hence, given the supporting literature and our robustness check, there is conclusive evidence that there are causal pathways between faraway basins. We have added these points in the discussion and methods section of the manuscript. (Lines 386-391 and 553-557)

At this point, we would also like to stress that this study's intended purpose is not to demonstrate the exact links between river basins but to demonstrate the perturbation of the Indian monsoon by river interlinking via land-atmosphere feedback. Such possibilities have not been considered in the impact assessment studies of large-scale hydrological projects like river interlinking. We have now explicitly mentioned this point in the manuscript discussion section. (Lines 419-423)

We acknowledge that we did not discuss the stationarity of the dataset in the manuscript. However, since we have separately performed analysis for each year, the non-stationarity of processes, if present, would have resulted in temporally changing links for each year instead of spurious links since each year is expected to be stationary. These temporally changing links would have been filtered out in our second step of the robustness check mentioned above (links are considered robust only if appearing more than 50% of the time). Regardless, we have now tested our dataset for stationarity using the Augmented Dickey-Fuller (ADF) test (<https://doi.org/10.2307/2286348>) and found that all variables in our dataset are stationary with p -value < 0.01. We have now reported this in lines 448-450 of the methods section.

In the face of global warming, climate change, and rapidly changing LULC, there is no denying that these links will change in the future. Hence, this study does not intend to convey exact pathways between river basins; rather, we intend to demonstrate the perturbation of large-scale flow by large-scale projects like river interlinking.

The points raised by the reviewer have helped us learn a lot and significantly improved the quality of the manuscript. We are grateful to the reviewer for providing insightful comments.

REVIEWERS' COMMENTS

Reviewer #2 (Remarks to the Author):

All my previous comments have been well addressed. I have no further comment.